# Interannual variations in the seasonal cyle of extreme precipitation in Germany and the response to climate change

Madlen Peter[1], Henning W. Rust[1], and Uwe Ulbrich[1]

[1]Institute of Meteorology, Freie Universität Berlin, Carl-Heinrich-Becker-Weg 6-10, 12165 Berlin, Germany

**Correspondence:** Madlen Peter (madlen.peter@met.fu-berlin.de)

**Abstract.** Annual maxima of daily precipitation sums can be typically described well with a stationary generalized extreme value (GEV) distribution. In many regions of the world, such a description does also work well for monthly maxima for a given month of the year. However, the description of seasonal and interannual variations requires the use of non-stationary models. Therefore in this paper we propose a non-stationary modelling strategy applied to long time series from rain gauges in Germany. Seasonal variations in the GEV parameters are modelled with a series of harmonic functions and interannual variations with higher ordered orthogonal polynomials. By including interactions between the terms, we allow for the seasonal cycle to change with time. Frequently, the shape parameter $\xi$ of the GEV is estimated as a constant value also in otherwise instationary models. Here, we allow for seasonal-interannual variations and find that this is benefical. A suitable model for each time series is selected with a step-wise forward regression method using the Bayesian Information Criterion (BIC). A cross-validated verification with the Quantile Skill Score (QSS) and its decomposition reveals a performance gain of seasonal-interannual varying return levels with respect to a model allowing for seasonal variations only. Some evidence can be found that the impact of climate change on extreme precipitation in Germany can be detected, whereas changes are regionally very different. In general an increase of return levels is more prevalent than a decrease. The median of the extreme precipitation distribution (2-year return level) generally increases during spring and autumn and is shifted to later times in the year, heavy precipitation (100-year return level) rises mainly in summer and occurs earlier in the year.

## 1 Introduction

Climate Change has been identified as the cause of increasing risks from meteorological extreme events affecting almost all areas of economy, nature and human life and and those will be even more endangered in the future (Pörtner et al., 2022, and the references therein). One of the main targets of current and future generations is to avoid further changes and to develop adaptation strategies to reduce risks and burdens.

While climate change can be measured very reliably for the surface temperature, for other variables like extreme precipitation the connection is not yet clear. For regions with good data availability, it has already been shown that frequency and intensity of heavy precipitation have likely increased on the global scale (Wehner et al., 2021). Furthermore, climate projections show that future extreme precipitation will continue to intensify (e.g. Pörtner et al., 2022; Rajczak et al., 2013). Since the consequences of heavy precipitation are extensive and can lead to different threats and damages, for example, due to flash floods, river floodings,

mudslides or soil erosion, an accurate assessment of extreme precipitation changes is crucial for an adequate adaptation. The potential risk due to extreme precipitation is not only dependent on its magnitude, but it also can be related to a change in its seasonal cycle. For example, a shift of strong precipitation from summer to spring lead to an increased flood risk due to a larger likelihood of strong rainfall and snow melt occuring at the same time (Vormoor et al., 2015; Teegavarapu, 2012). Furthermore, crop losses may rise, since plants are more vulnerable during earlier growing stages (Rosenzweig et al., 2002; Zeppel et al., 2014; Derbile and Kasei, 2012).

Analyses of extreme precipitation in Germany for different seasons have already been done (Zolina et al., 2008; Łupikasza, 2017; Fischer et al., 2018, 2019; Zeder and Fischer, 2020; Ulrich et al., 2021). Zolina et al. (2008) and Łupikasza (2017) analysed quantiles of daily precipitation sums separately for the seasons DJF, MAM, JJA and SON, while Fischer et al. (2018, 2019) used available data more efficiently by modelling monthly maxima of daily precipitation sums for all months simultaneously. This approach has been proven to lead to more robust and reliable results than considering months separately. Ulrich et al. (2021) extended this method by including different durations to efficiently estimate intensity-duration-frequency curves. Furthermore, Zeder and Fischer (2020) analysed the effect of climate change on seasonal extreme precipitation and found a positive connection to the north-hemispheric temperature rise. In our approach we combine the simultaneous modelling of available data for all months with interannual variations, thus accounting for potential changes of the seasonality due to climate change and natural variability. Here, we point out that when referring to interannual variations, we are not addressing differences between successive years, but rather the trend over the entire observation period, which could be potentially non-linear.

Extreme value statistics (EVS) (e.g. Coles, 2001; Bousquet and Bernardara, 2021) is used to quantify the magnitude and occurrence probabilities of these seasonal-interannually varying extremes. EVS have been applied in many different research fields (e.g. Katz et al., 2002; Ferreira et al., 2017; Szigeti et al., 2020; Arun et al., 2022). One way to analyse extremes is the block maxima approach, where the observations are divided into blocks with equal lengths. The probability distribution for the maxima of these blocks is represented by the Generalized Extreme Value (GEV) distribution. Instead of considering annual maxima of precipitation, which are frequently used in risk assessment, we take a monthly block size to resolve the seasonal cyle. Contrary to a stationary approach with an individual extreme value model for each calendar month, we take advantage of the smooth variations in the probability distributions of the block maxima across adjacent calendar months. Because of the periodic nature of the seasonal changes a series of harmonic functions is an appropriate choice for describing the corresponding variations in the GEV parameters. This modelling strategy has already been widely applied (e.g. Méndez et al., 2007; Rust et al., 2009; Galiatsatou and Prinos, 2014; Fischer et al., 2019; Min and Halim, 2020). It has been shown to provide more accurate monthly and annual return levels (quantiles of the GEV) (Fischer et al., 2018).

Interannual variations in precipitation have been shown to be associated with its natural variability (e.g. Willems, 2013), increased air temperatures (Trenberth et al., 2003; Westra et al., 2013, 2014) and other effects influencing large-scale atmospheric circulations and precipitation characteristics (Pinto et al., 2007, 2009; Davini and d'Andrea, 2020; Detring et al., 2021). Most of these effects are highly non-linear and their roles are difficult to quantify. Here, we use time as proxy to combine those different unknown effects. One possibility to model non-linear interannual changes is polynomial regression (e.g. Kjesbu et al., 1998; Mudelsee, 2019; Bahrami and Mahmoudi, 2022). Orthogonal polynomials are used to reduce multicollinearity and to

improve the parameter estimation (Shacham and Brauner, 1997). Here, we use Legendre polynomials up to an order of five to describe the variations across years. This enables on the one hand the reflection of changes potentially associated with climate change and on the other hand allows for modelling of natural variability in extreme precipitation. The concept of using higher-ordered Legendre Polynomials has also been applied to assess spatial variations (Ambrosino et al., 2011; Rust et al., 2013; Fischer et al., 2019). As the seasonal and interannual covariates are conceptionally equal, we combine both approaches. Additionally, interactions between the covariates allow the seasonal cycle to change across years.

The goal of this paper is to assess the performance of the seasonal-interannual modelling with a special attention to a flexible shape parameter $\xi$. This parameter is difficult to estimate as it interferes with the scale parameter (Ribereau et al., 2011) and requires long records for reliable results (Papalexiou and Koutsoyiannis, 2013). Nevertheless, it describes the behaviour of the very rare events and consequently plays an important role for assessing extreme precipiation changes. Furthermore, the possible impact of climate change on the seasonal cycle of extreme precipitation is analysed. We formulate three research questions to be addressed in this study:

**RQ1** Can a model with interannual variations better represent the observations than a seasonal-only model?

**RQ2** How important is a flexible shape parameter to reflect recorded variations?

**RQ3** How does climate change affect the seasonal cycle of extreme precipitation in Germany?

We carry out this investigation for observations from Germany with more than 500 long ($\geq 80$ years) records, presented in Sec. 2. The seasonal-interannual modelling is described in Sec. 3. Model selection and validation tools are covered in Sec. 4. The gain of modelling seasonal-interannual variations with respect to a just seasonal model (RQ1) and the importance of a flexible shape parameter $\xi$ (RQ2) are assessed in Sec. 5 and Sec. 6. The impact of climate change on the seasonal cycle of heavy precipitation (RQ3) is tackled in Sec. 7. Finally, we discuss the results in Sec. 8.

## 2   Data

A dataset of almost 5700 rain gauges measuring daily precipitation amounts (DWD, 2021) is provided by the German Meteorological Service (Deutscher Wetterdienst, DWD) via the continously updated 'open-data-server' (DWD, 2022). Those observation stations are set up according to the WMO guidelines (WMO, 1996). The daily sums of precipitation are obtained from amounts accumulated between 5:50 UTC to 5:50 UTC of the following day and have been checked for spatial consistency DWD (2021).

For investigating long term trends a sufficently long time series is crucial, thus we only consider the most recent stations with at least 80 years of observations lasting until 2021-12-31. We allow for missing values and larger gaps of several consecutive years, often occurring for the years of the second world war. The 519 stations fullfilling the mentioned criteria are depicted in Fig. 1, the colour coding shows the station's altitude. The locations are not homogenously distributed in space: some areas are closely covered, while for other areas, for example, in the East of Germany or in the State of Saarland in western Germany, long time records which are still being updated are missing. The common time period for all 519 observation records covers

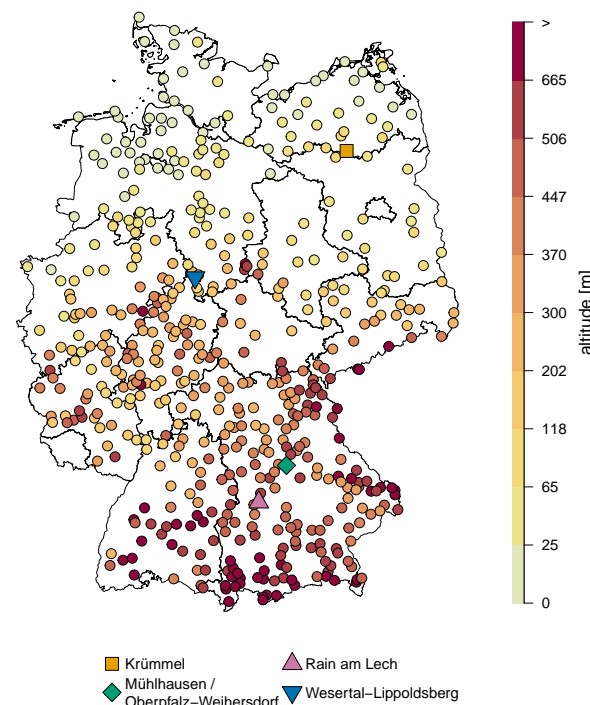

**Figure 1.** 519 long stations covering at least the years from 1941 to 2021. Station altitude [m] is encode with colours. Additionally, the locations of stations *Krümmel* (orange rectangle), *Mühlhausen / Oberpfalz-Weihersdorf* (green rhombus), *Rain am Lech* (violet pointing up triangle) and *Wesertal-Lippoldsberg* (blue pointing down triangle) are depicted.

the years from 1941 to 2021. The four stations *Krümmel* (1899-01-01 until 2021-12-31), *Mühlhausen / Oberpfalz-Weiherdorf* (1931-01-01 until 2021-12-31), *Rain am Lech* (1899-01-01 until 2021-12-31) and *Wesertal-Lippoldsberg* (1931-01-01 until 95 2021-12-31) are highlighted in Fig. 1 and will be discussed exemplarily in this study. We have selected these stations as they are characterised by different changes in seasonality (see Sec. 7) represented by divergent model setups (see Sec. 4.1). Additionally, their interannual changes are more pronounced than for other stations. We consider monthly maxima of daily precipitation sums while months with less than 27 measured days are discarded from the analysis.

## 3 Modelling seasonal-interannual extreme precipitation

In order to describe the changes in seasonality of extreme precipitation, we build a statistical model. This can be done with concepts of extreme value statistics (EVS), which are widely explored and applied in different scientific fields (e.g., for the financial sector (Gilli et al., 2006; Gkillas and Katsiampa, 2018); or for geosciences (Yiou et al., 2006; Naveau et al., 2005; Ulrich et al., 2020; Fauer et al., 2021; Moghaddasi et al., 2022; Jurado et al., 2022)). One major strategy in EVS is the block-

maxima approach leading to an asymptotic model for extreme values: the Generalized Extreme Value (GEV) distribution, briefly described in the following.

## 3.1 Block-maxima approach

For a sequence of independent and identically distributed (iid) random variables $X_1, ..., X_n$ the block maxima are defined as

$$M_n = \max\{X_1, \ldots, X_n\}. \tag{1}$$

The Fisher–Tippett–Gnedenko theorem (FTGT) (Coles, 2001) states, that for a suffiently large block size $n$, the probability distribution function (PDF) of the block maxima can be well described either with the Gumbel-, the Fréchet, or the Weibull distribution. The three families can be combined into the General Extreme Value (GEV) distribution

$$G(z) = \begin{cases} \exp\left\{-\left[1 + \xi\left(\frac{z-\mu}{\sigma}\right)\right]^{-1/\xi}\right\} & ,\xi \neq 0 \\ \exp\left[-\exp\left\{-\left(\frac{z-\mu}{\sigma}\right)\right\}\right] & ,\xi = 0 \end{cases} \tag{2}$$

with $\{z : 1 + \xi(z - \mu)/\sigma > 0\}$. This distribution has three parameters: location $-\infty < \mu < \infty$, specifying the position of the PDF, scale $\sigma > 0$ defining the width of the PDF and shape $-\infty < \xi < \infty$ characterizing the behaviour of the upper tail. The value of $\xi$ determines the type of extreme value distributions ($\lim \xi \to 0$: Gumbel, $\xi > 0$: Fréchet, $\xi < 0$: Weibull).

The choice of the appropriate block size is dependent on the nature of the considered random variable (Embrechts et al., 1997; Rust, 2009). Studies (e.g. Rust et al., 2009; Maraun et al., 2009) show, that a block size of one month is already sufficiently large for extreme precipitation in the mid-latitudes. Others confirm the choice of monthly maxima for the considered datasets by using Q-Q-Plots (Fischer et al., 2018, 2019). The advantage of a higher temporal resolution of the maxima series makes it possible to analyse the seasonal cycle of extreme precipitation. This requires independent block maxima of successive months. However, this assumption can be violated if two monthly maxima belong to the same precipitation event, e.g. if one maximum occurs at the end of the month and the second one at the beginning of the next month. For the given records, about 0.6% of the monthly maxima have been registered at successive days. Since the percentage is low, we neglect temporal dependances and assume independent monthly maxima.

In the frame idea of vector generalized linear models (VGLM Yee, 2015b), we describe the variation of GEV parameters as linear functions depending on different variables. The variations throughout the course of the year are captured in Fischer et al. (2018) and are extended to a seasonal-spatial variation of extreme precipitation in Fischer et al. (2019). Ulrich et al. (2020) applied spatial variations to a duration-dependant GEV. Additionally, a change in the GEV parameters with other meteorological variables, e.g. temperature and the El Niño-Southern Oscillation (ENSO) index (Villafuerte et al., 2015), the North Atlantic Oscillation (NAO) index (Golroudbary et al., 2016) or an index of synoptic airflow (Maraun et al., 2011)) have been accomplished by various authors. In this study the seasonal and interannual variations are in focus. For each of the three GEV parameters, we build a linear model as shown here in a conceptual way for $\mu$:

$$g(\mu) = \mu_0 + \sum \mu_i X_i + \sum \mu_{i,j} X_i X_j, \tag{3}$$

where $g$ is a link function – for $\mu$ the identity function $g(\mu) = \mu$, for $\sigma$ the logarithm $g(\sigma) = \ln(\sigma)$ and for $\xi$ the logarithm with an offset of 0.5 $g(\xi) = \ln(\xi) + 0.5 - \mu_0$ denotes the constant intercept (offset), the second term the direct effects of a covariate $X_i$, e.g. seasonal or interannual, and the third term the interactions between different dimensions (indicated by $i$ and $j$), e.g. seasonal and interannual.

## 3.2 Modelling seasonality

To account for the periodic nature of the seasonal cycle, the dependence of GEV parameters on the months can be described with a series of harmonic oscillations with amplitude $A$ and a phase $\alpha$. For the first harmonic oscillation (h=1) the location parameter $\mu$ can be written as

$$\mu_{c_t}^{h=1} = \mu_0 + A\sin(\omega c_t + \alpha), \tag{4}$$

with $t = 1, \ldots, 12$ the months in the year, $c_t$ the centre of the $t$-th month given in days starting from January $1^{\text{st}}$ and $\omega = 2\pi/365.25$ the angular frequency of earth's rotation.

To describe the oscillation Eq. 4 in the framework of a linear model, we use a linear combination of sine and cosine

$$\mu_{c_t}^{h=1} = \mu_0 + a\sin(\omega c_t) + b\cos(\omega c_t), \tag{5}$$

with the coefficients $a$ and $b$ defining the amplitude $A$ and the phase $\alpha$ as

$$A = \sqrt{a^2 + b^2} \tag{6}$$

and

$$\alpha = \begin{cases} \frac{\pi}{2}, & a = 0 \\ 0, & b = 0 \\ \text{atan2}(b, a), & a \neq 0, b \neq 0 \end{cases} \tag{7}$$

The harmonic series for location

$$\mu_{c_t} = \mu_0 + \sum_{h=1}^{H} (\mu_{h_{\sin}} \sin(h\omega c_t) + \mu_{h_{\cos}} \cos(h\omega c_t)), \tag{8}$$

with $h = 1, \ldots, H$ indicating the order of harmonic function, approximates an arbitrary periodic function (Priestley, 1992).

## 3.3 Modelling interannual variation

To capture interannual variations, polynomials typically provide a good approximation. With orthogonal polynomials such as Legendre polynomials, we avoid dependence between terms which has been proven useful for modelling spatial variations (Rust et al., 2013; Fischer et al., 2019). We adopt this approach here to describe interannual variations. For the location

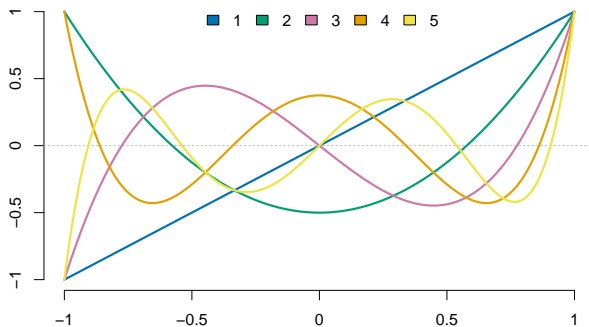

**Figure 2.** The Legendre polynomials for the orders one to five.

parameter $\mu$ this reads

$$\mu_Y = \sum_{i=1}^{I} \mu_{i_P} P_i(Y),\tag{9}$$

with $i = 1,\dots,I$ indicating the order of Legendre polynomial $P$ and $Y$ the transformed year of the observation. The transformation of the time axis needs to be done since Legendre polynomials are only defined on $[-1,1]$. For that we use

$$Y = \frac{2(Y' - Y'_{min})}{Y'_{max} - Y'_{min}} - 1,\tag{10}$$

with $Y'$ being the respective year and $Y'_{min}$ / $Y'_{max}$ denoting the first / last year of the record. This transformation has been done for each station separately depending on its observation period. We exemplify the Legendre polynomials up to order five in Fig. 2.

### 3.4 Modelling the interannual variation of seasonality

We focus on the interannual changes of the seasonal cycle, which can be incorporated into the statistical model using interactions between the seasonal and interannual terms in the predictor of the vector generalized linear model. It can be thought of as amplitude and phase of the seasonal cycle changing in time.

Using Eq. 6 with coefficients $a(Y) = a\,P_i(Y)$ and $b(Y) = b\,P_i(Y)$ being modulated by time dependent Legendre polynomials $P(Y)$, the amplitude $A$ varies with the square of the Legendre polynomials $P(Y)$: with the compact support on $[-1,1]$ interaction of harmonics with a linear change with years $P_1(Y)$ leads to a quadratic change for the squared amplitude $A^2$ and thus to an interaction term $a\,\sin(\omega\,c_t)\,P_1(Y) + b\,\cos(\omega\,c_t)\,P_1(Y)$ with decreasing amplitude on $[-1,0]$ and increasing amplitude on $[0,1]$ as illustrated in Fig. 3 (top row, with $b = 0$ for simplicity). The following rows of Fig. 3 show the corresponding interaction terms for $P_2(Y)$ (middle) and $P_3(Y)$ (bottom).

To avoid the bipartite behaviour we use two transformations of the Legendre polynomials: $P_i^- = 1/2\,(P_i(Y) - 1)$ and $P_i^+ = 1/2\,(P_i(Y) + 1)$, such that $P_i^{t1} = 1/2\,(P_i(Y) - 1) : [-1,1] \to [-1,0]$ and $P_i^{t2} = 1/2\,(P_i(Y) + 1) : [-1,1] \to [0,1]$.

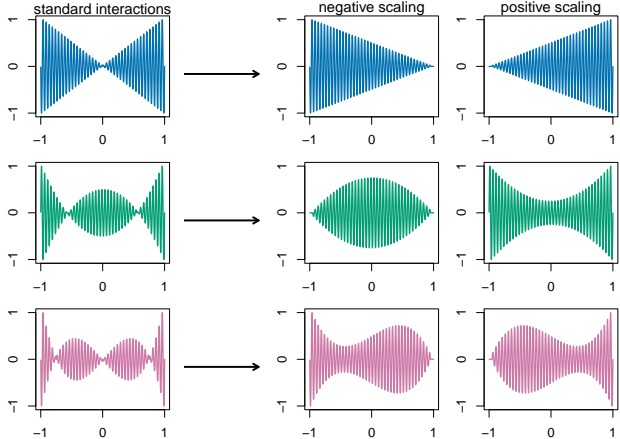

**Figure 3.** Standard interaction terms (left-hand side) between the first order sine and the Legendre polynomials of order one (top row, orange), order two (mid row, green) and order three (bottom row, red). A negative and a positive scaling of the Legendre polynomials lead to the desired interaction terms (right-hand side).

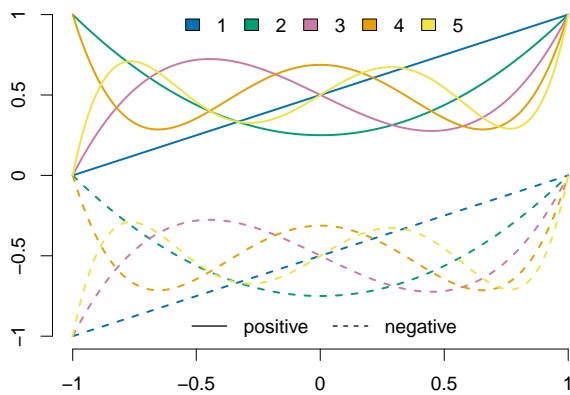

**Figure 4.** Positively (solid) and negatively (dashed) transformed Legendre polynomials of order one to five.

The transformed Legendre polynomials are illustrated in Fig. 4.

Thus, the interactions with the harmonic functions for the location parameter $\mu$ can be expressed as

$$
\mu_{\mathrm{int}}^{-} = \sum_{h=1}^{H} \sum_{i=1}^{I} \Big( \mu_{h,i,\sin}^{-} \sin(h\,\omega\,c_t)\, \frac{P_i(Y)-1}{2} + \mu_{h,i,\cos}^{-} \cos(h\,\omega\,c_t)\, \frac{P_i(Y)-1}{2} \Big)
$$

(11)

$$\mu_{\text{int}}^{+} = \sum_{h=1}^{H} \sum_{i=1}^{I} \left( \mu_{h,i,\sin}^{+} \sin(h\,\omega\,c_t) \frac{P_i(Y)+1}{2} + \right.$$
$$\left. \mu_{h,i,\cos}^{+} \cos(h\,\omega\,c_t) \frac{P_i(Y)+1}{2} \right) \tag{12}$$

These terms show the desired behaviour as depicted exemplarily in Fig. 3 on the right-hand side.

Combining the seasonal and interannual variations with these interactions leads to a flexible model for the location parameter

$$g(\mu) = \mu_0 + \mu_{c_t} + \mu_Y + \mu_{\text{int}}^{-} + \mu_{\text{int}}^{+}. \tag{13}$$

Using a VGLM, we allow the scale $\sigma$ and shape $\xi$ to vary in the same way. In many publications (e.g., Golroudbary et al., 2016; Rust et al., 2009) the shape parameter is described merely with a constant offset $\xi_0$ to be estimated or is even set to a fixed value. The reason is that this parameter is regarded as difficult to estimate as it describes the behaviour of the most extreme and thus very rare events. Papalexiou and Koutsoyiannis (2013) state that *"the record length strongly affects the estimate of the GEV shape parameter and long records are needed for reliable estimates."*. We assume our dataset will be sufficently long. Fischer et al. (2019) have illustrated by means of an example station in Germany, that a pronounced seasonal cycle in $\xi$ exists with lower, partly negative, values in winter and higher values in summer. Thoses differences could be explained with the predominance of less intense stratiforme precipitation in the winter months, and more intense convective precipitation in the summer months. The performance gain of a seasonal-interannual shape parameter will be discussed in detail in Sec. 6.

### 3.5 Return levels

The $p$-quantile of the GEV gives the return level $r_p$ for a certain non-exceedance probability $p$ (or occurrence probability $1-p$) and can be written as

$$r_p = \mu - \frac{\sigma}{\xi} [1 - [-log(p)]^{-\xi}] \tag{14}$$

Instead of stating the non-exceedance probabilities it is common to consider the respective average return period $T = \frac{1}{1-p}$. The interpretation is that the return level is exceeded on average once in this particular time period. Since we consider return levels changing in time, the concept of a temporal change of a return period might be difficult to capture. Thus, in the following we refer to a (time dependent) non-exceedance probability $p(t)$ and the return period $T(t)$ simultaneously. As we consider monthly maxima we calculate as well monthly return levels. Similar to e.g. 100-year return levels obtained with annual maxima, we determine the 100-January return levels, the 100-February return levels, and so on. In the following we state them as monthly 100-year return levels instead of naming respective months. This should not be confused with annual return levels. However, they can be calculated as well with monthly maxima leading to more accurate and reliable annual results (Maraun et al., 2009; Fischer et al., 2018).

## 4 Model Building and Verification

### 4.1 Stepwise Model Selection

After introducing the model setup in the previous Section, the maximum order for harmonic functions and Legendre polynomials have to be selected. Here, we set maximum orders $H$ and $I$ to five to ensure a feasible model selection procedure. Considering the data and the selected models didnot show evidence for an additional including of higher orders than five. With $H = I = 5$ the full model consist of 348 coefficients (116 for each GEV parameter: 1 constant offset, 10 for seasonal variations, 5 for interannual variations, 100 interaction terms) for each station separately. This model is reduced to the necessary complexity with stepwise model selection using the Bayesian Information Criterion (BIC) (e.g. Neath and Cavanaugh, 2012). The procedure has two parts: first, only the direct effects are selected; in the second part, the interactions are added subsequently. Starting point is the stationary GEV (Eq. 2). In each iteration, every possible covariate is added once to the reference (in the first iteration: stationary GEV) and the BIC is determined. For the first iteration of part one this leads to 45 different models (15 for each GEV parameter). The model with the lowest BIC is selected as the best candidate for the next step. If the difference $\Delta \mathrm{BIC} = \mathrm{BIC}_{\mathrm{ref}} - \mathrm{BIC}_{\mathrm{model}} > 2$ (as suggested by Fabozzi et al., 2014) the model is considered superior to the reference and becomes the new reference for the next iteration. Again, all remaining covariates are probed once for model improvement (leading to 44 different models for iteration two of part one) and the stepwise model selection is continued. If $\Delta \mathrm{BIC} < 2$ for all covariates to probe, the current reference model is taken as the final model.

Now, the procedure is repeated for interaction terms starting with the final model from part one. As we are interested in the gain of including interannual variations (RQ1), we select for each station a seasonal model without interannual variability as references in to use for RQ1, Sec. 5). To address RQ2, model selection with different setups are used; details are given in Sec. 6.

Fig. 5 illustrates the stepwise selection for the four example stations. The BIC (x-axis) decreases with adding necessary covariates (y-axis from top to bottom). All covariates listed in the panels (Fig. 5) for the corresponding parameter (color) are included in the final model. Numbers following *sin*, *cos* and *P* indicate the respective harmonic/polynomial order. Terms with a colon denote interactions. For all four example stations interannual terms in addition to seasonal ones have been included following the procedure described above; however, the type of interannual variation differs for each station. On one hand, stations *Rain am Lech* and *Wesertal-Lippoldsberg* are characterized by a linear change in parameters. These changes occur in the $\mu$ and $\sigma$ parameters for the former station, while for the latter the seasonal cycle of $\xi$ changes linearly with the years. On the other hand the extreme precipitation of the stations *Mühlhausen / Oberpfalz-Weihersdorf* and *Krümmel* are described with higher order Legendre Polynomials.

The model selection procedure was applied for all 519 stations individually. About 65% of the stations (338/519) prefer a model *with* an interannual component. Those gauge stations are roughly equally distributed in space and no common characteristics (e.g. stations altitude or record length) are apparent compared to stations *without* an interannual component. All models of the 338 stations contain as well seasonal variations. The properties of the interannual variability of those models are depicted in Fig. 6: (a) indicates those GEV parameters which show an interannual component; (b) shows whether an interannual com-

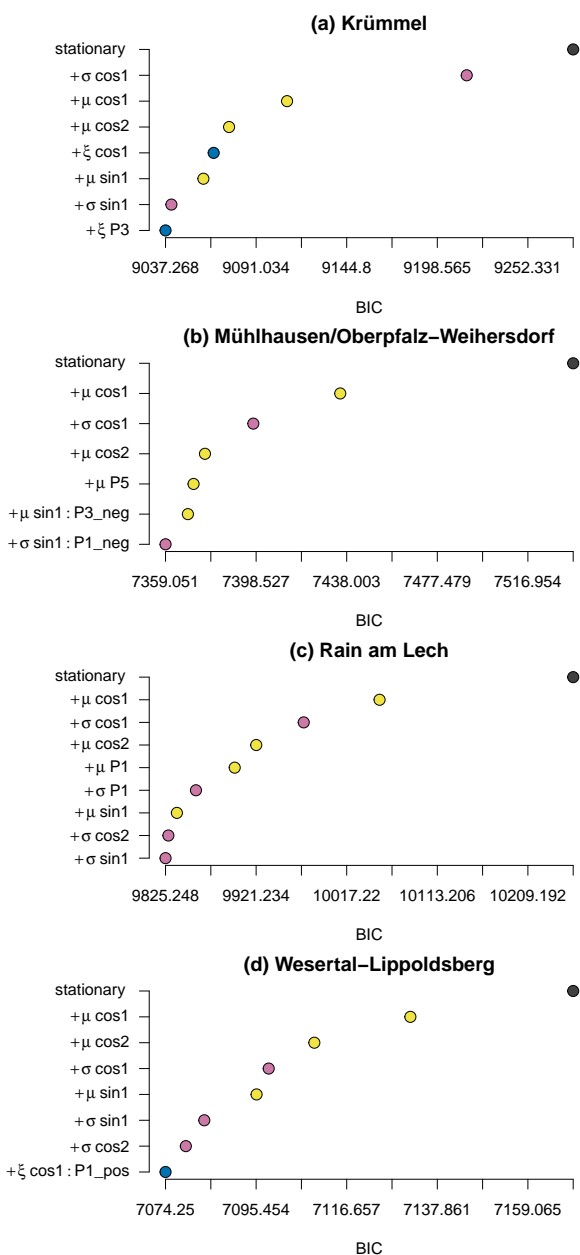

**Figure 5.** Stepwise model selection for example stations. BIC against stepwise selected covariates for location (yellow), scale (pink) and shape (blue) for each iterations. All covariates listed are included in the final model.

ponent is part of a direct effect and/or an interaction; (c) gives the counts and portions of the selected Legendre Polynomials (x-axis) for the GEV parameter (y-axis) and kind of covariate (direkt: top, interactions: bottom). We do not show the spatial distribution of the Legendre Polynomials since no clear pattern can be detected.

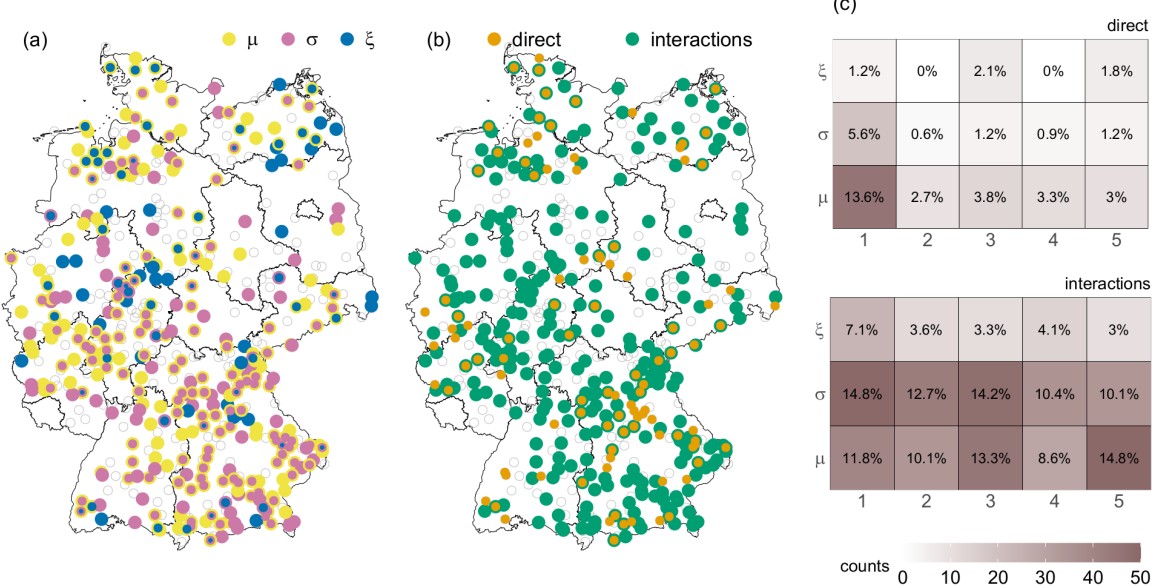

**Figure 6.** Properties of the interannual variability components of those 338 models including at least one. (a) GEV parameter with interannual component, location (yellow), scale (pink) and shape (blue). (b): direct (orange) and/or interactions (green) with interannual components. (c): counts (color intensity) and portion (percentage) of selected orders of the Legendre Polynomials (x-axis) for different GEV parameters (y-axis) divided for direct (top) und interactions (bottom) for the 338 models. Stations with no interannual component are marked as transparent circles.

It can be seen that the selected interannual covariates are partly very variable in space. This can be explained by 1) a large spatial variability in extreme precipitation due to partly small-scaled events and 2) the model selection procedure, which chooses one suitable model, even if other models are comparably appropriate. However, common characteristics can be detected: The GEV's location and scale parameter are mainly affected and interannual changes of the seasonal cycle (interactions) dominates. Nevertheless, changes of the shape parameter and changes without affecting the seasonal behaviour occur, often for several stations of the same region, indicating common local characteristics. The stations with direct effects are mainly characterized by a linear interannual change in the location parameter. For the interactions, the prefered Legendre polynomial is not so obvious.

## 4.2 Model Verification Tools

To answer RQ1 and RQ2 the performance of a model with respect to a reference has to be analysed. We use the Quantile Skill Score (QSS) (Bentzien and Friederichs, 2014; Friederichs and Hense, 2007), which is based on the Quantile Score (QS) defined as

$$QS = \frac{1}{N} \sum_{n=1}^{N} \rho_p (o_n - r_{p,n}). \tag{15}$$

Here $\rho_p$ denotes the check-function, defined as

$$\rho_p(u) = \begin{cases} pu & u \geq 0 \\ (p-1)u & u < 0 \end{cases} \tag{16}$$

with $u = o_n - r_{p,n}$. The QS is a weighted mean of differences between the $N$ observations $o_n$ and the quantiles (return levels) $r_{p,n}$ for a certain non-exceedance probability $p$. It is positively oriented and optimal at zero. We use leave-one-year-out cross validation to obtain a robust Quantile Score estimate.

The Quantile Skill Score (QSS) is defined as

$$\text{QSS} = \frac{\text{QS}_{model} - \text{QS}_{ref}}{\text{QS}_{perf} - \text{QS}_{ref}} = 1 - \frac{\text{QS}_{model}}{\text{QS}_{ref}}, \tag{17}$$

with the perfect score $\text{QS}_{perf} = 0$. For a model outperforming the reference, the QSS is in the range $(0, 1]$ giving the fraction of improvement with respect to the difference between the perfect and the reference model; for models worse than the reference, $\text{QSS} < 0$.

For stratifying verification along months or stations we use the decomposition of the QSS (Richling et al., in preparation) to learn about pecularities of certain subset,

$$\text{QSS} = \sum_i^K \frac{N_i}{N} \cdot \left( 1 - \frac{\text{QS}_{i,model}}{\text{QS}_{i,ref}} \right) \cdot \frac{\text{QS}_{i,ref}}{\text{QS}_{ref}}, \tag{18}$$

with $K$ being the number of different subsets, e.g. for monthly stratification $K = 12$. The Quantile Skill Score for the full data set can be decomposed into the sum of a weighted Quantile Skill Score for the different subsets. The term $1 - \frac{\text{QS}_{i,model}}{\text{QS}_{i,ref}}$ in Eq. 18 represents the subset QSS, weighted on the one hand with the so-called *frequency weighting* $\frac{N_i}{N}$, indicating how many data points can be attributed to that subset, and on the other hand with the *reference weighting* $\frac{\text{QS}_{i,ref}}{\text{QS}_{ref}}$, indicating how well the reference can represent the data for the given subsets with respect to the complete dataset. The weighted subset QSS can be regarded as the contributions to the total QSS.

## 5  Gain of interannual variability

We address RQ1: Can a model with interannual variations better represent the observations than a seasonal-only model? As mentioned in Sec. 4.1, only for 338 of 519 stations ($\sim 65\%$) a model with at least one interannual component in any of the GEV parameter was chosen. To assess the importance of the interannual variations of these 338 stations we analyse the skill with respect to the seasonal-only model. Tab. 1 shows the total QSS for different non-exceedance probabilities (return periods). Skill is positive but small $\lesssim 2\%$, increasing with non-exceedance probability (return period). The latter has to be interpreted with care as there are very few observations in the range of the upper quantiles. Return levels with a return period higher than the time range of the data should be treated cautiously, since the quantile score can not reasonably evaluate those values (Fauer and Rust, 2023). As we consider for each station at least 80 years of observations, this only matters for non-exceedance probabilities

| $p\,(T)$ | QSS |
|---|---|
| $0.5(2a)$ | 0.006 |
| $0.8(5a)$ | 0.007 |
| $0.9(10a)$ | 0.008 |
| $0.95(20a)$ | 0.010 |
| $0.9\bar{6}(30a)$ | 0.012 |
| $0.98(50a)$ | 0.015 |
| $0.99(100a)$ | 0.019 |
| $0.995(200a)$ | 0.021 |

**Table 1.** Total QSS for different non-exceedance probabilities $p$ (return periods $T$) of the seasonal-interannual model with respect to the seasonal-only model averaged over 338 stations with a interannual varying component.

(return periods) of 0.99 and 0.995 (100 and 200 years). The small increase in skill due to the inclusion of interannual variation is expected as most of the signal can be described already with the strong seasonal cycle.

We analyse whether interannual variations improve the estimates of return levels for a particular month and stratify the QSS along months, Fig. 7 (a). To unterstand the importance of the monthly subset scores, the reference weighting $QS_{i,ref}/QS_{ref}$ (b) and the contribution to the total skill score (c) are shown. The frequency weighting $N_i/N$ is (almost) identical for all subsets (as the records generally contain complete years) and is not shown. Averaged over all 338 stations the monthly QSS (a) is positive for all months and non-exceedance probabilities with spring (March, April) and summer (July, August) standing out. For the contribution to the total QSS, the reference weighting (b) gives more importance to the summer months, leading to the strongest constribution to total QSS in July. The structure of the reference weigthing term (b) indicates that the seasonal-only model does not represent the observations in summer as good as in winter. This probably indicates a stronger need for taking interannually varying return levels in summer into account. Adding the values of Fig. 7 (c) by row lead to the values depicted in Tab. 1. The monthly QSS averaged over 338 stations leads to a consistently positive skill, but the performance varies strongly for the different stations.

We now stratify verification also along stations ($K = 338$) and average over all timesteps. The subset QSS for the non-exceedance probabilities of $p = 0.9, 0.98, 0.99, 0.995$ are plotted in Fig. 8 and the reference weighting is exemplarily illustrated for $p = 0.99$ (since the pattern are similar for all non-exceedance probabilities) in Fig. 9. The frequency weighting and the contribution to the total QSS are not shown, since the first one does not exhibit any spatial pattern and the last one does not visually distinguish from the figure of the subset QSS. For most of the stations the seasonal-interannual model can represent the observations better than the seasonal-only model.

Only for a few records and higher non-exceedance probabilities / return periods the variations with the years lead to more uncertain return levels, for example station *Wesertal-Lippoldsberg*. The monthly contribution to the QSS for this station is depicted in Fig. 10 (b). The negative skill mainly arises from overestimated return levels for the summer months, especially for the recent years (visually verified in Sec. 7) . This merely occurs for higher return periods due to the interannual varying shape

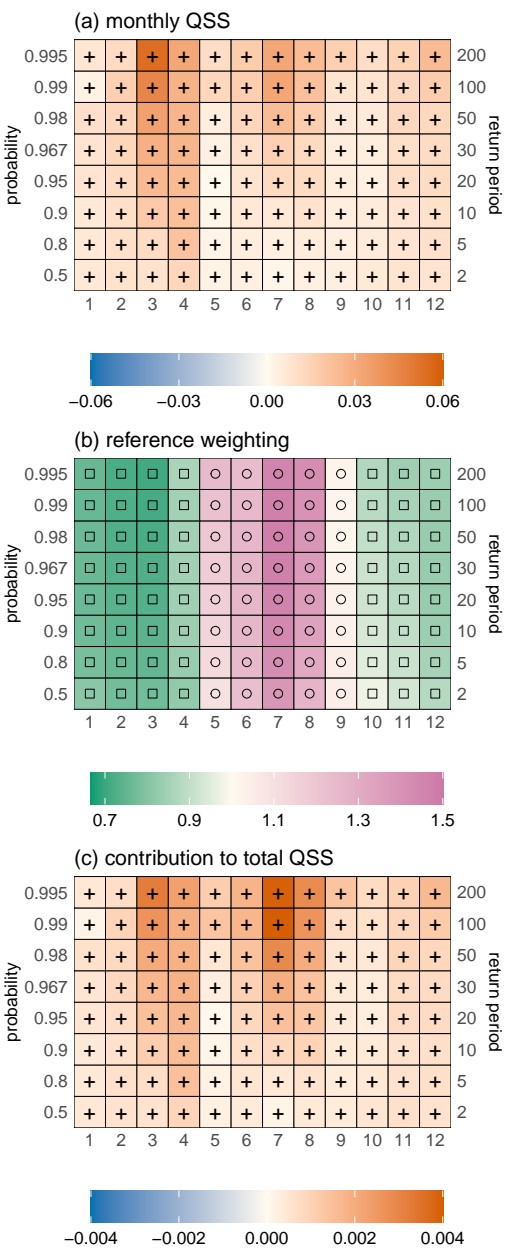

**Figure 7.** Subset QSS (a), reference weighting (b) and the contribution to the total QSS (weighted subset QSS) (c) for the months January to December (x-axis) averaged over 338 stations with interannual components for different non-exceedance probilities (left y-axis) / return periods (right y-axis). Positive / negative values (orange; plus sign / blue; minus sign) of the QSS (weighted QSS) indicate an increased / decreased performance of the seasonal-interannual model with respect to the seasonal-only model. The reference weighting describes how good the seasonal-only model describes the subset data with respect to the full dataset: green (squares) / pink marks (circles) a better/worse representation.

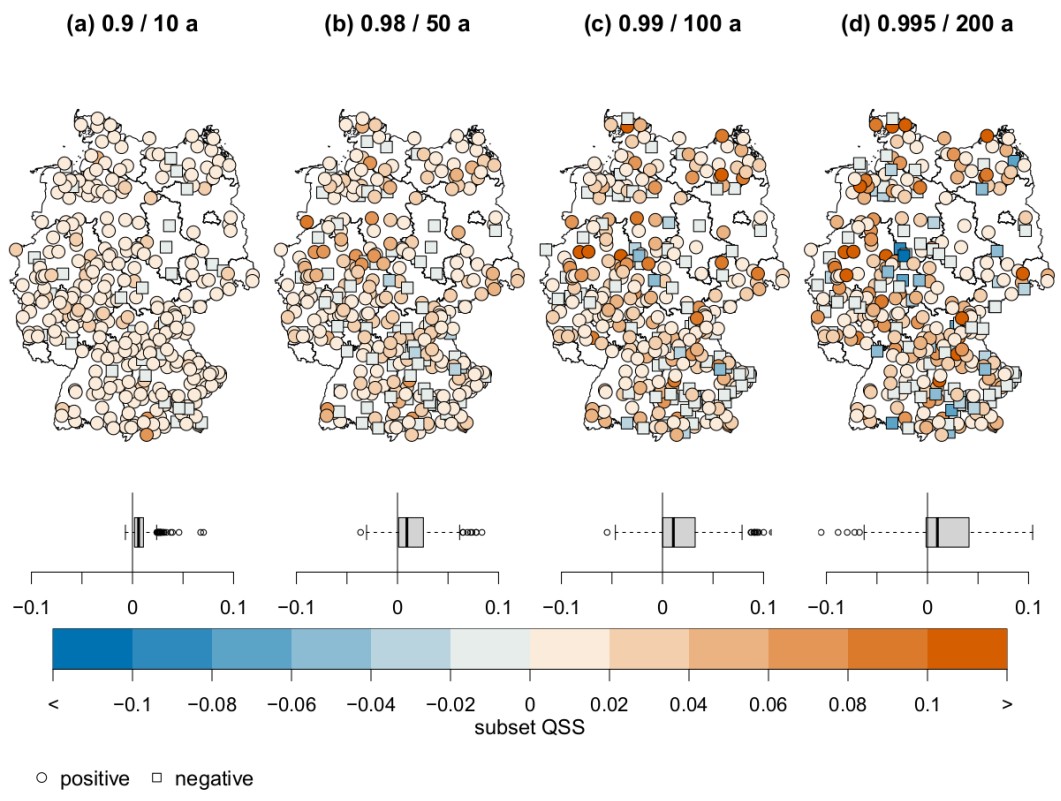

**Figure 8.** Subset QSS for 338 different stations for the non-exceedance probability / return period of (a) 0.9 / 10 years, (b) 0.98 / 50 years, (c) 0.99 / 100 years and (d) 0.995 / 200 years. The distribution of the subset QSS is depicted as Box-Whisker-Plot and in space (map). Positive / negative (orange circles / blue squares) values mark a gain / loss in skill.

parameter $\xi$ (see Fig. 6). However, a worse skill for stations with a interannual component in $\xi$ can not be detected in general (not shown). We discuss the change of the seasonal cycle of *Wesertal-Lippoldsberg* in more detail in Sec. 7.

Compared with the location heights of Fig. 1, the reference weighting for the station-wise analysis in Fig. 9 shows a clear relationship to stations altitude and specifies that the seasonal-only model can not reflect the data in mountainous regions as good as in lowlands. Analysing the skill of the seasonal-interannual model with respect to the altitude does not show an improvement especially for higher located stations (b, blue crosses). This might indicate that both model setups miss important mechanisms for extreme precipitation in mountainous regions (e.g. convection due to lifting or flow direction).

Thus, these processes can not be approximated by solely including temporal covariates but need to be modelled directly and/or via appropriate spatial covariates. This weak point can not be seen for the example stations, since they are located in the lowlands (*Krümmel*: 64 m, *Wesertal-Lippoldsberg*: 176 m) or in the low mountain ranges (*Rain am Lech*: 409 m, *Mühlhausen / Oberpfalz-Weihersdorf*: 420 m).

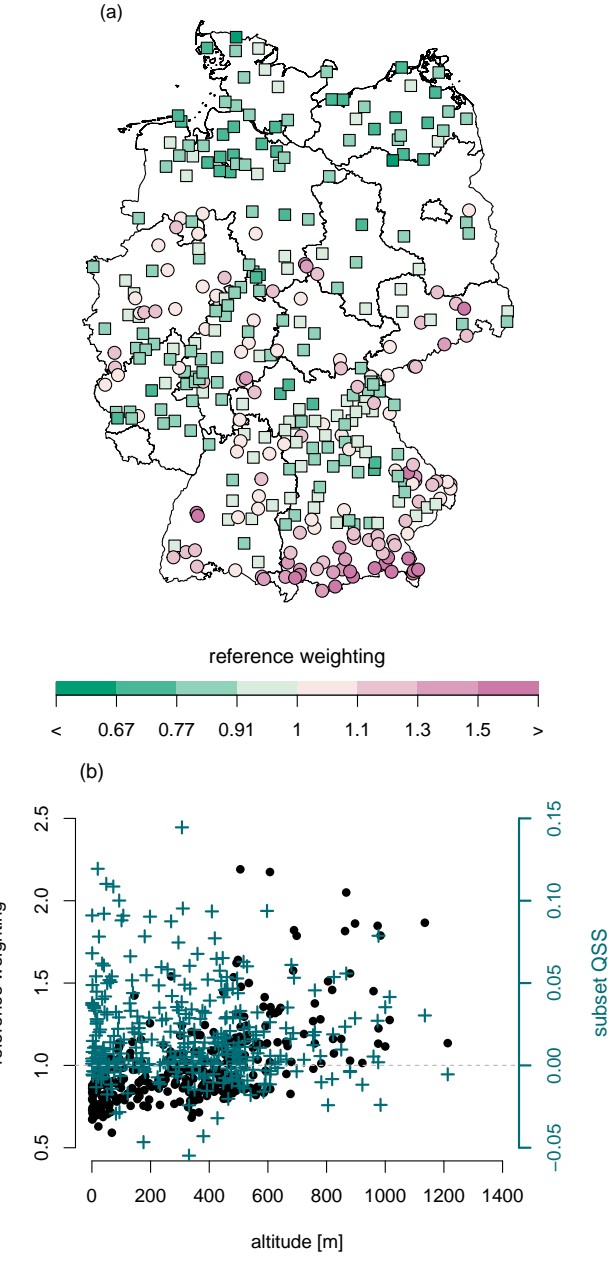

**Figure 9.** Map of reference weighting for non-exceedance probability 0.99 / return period 100 years (a). Green / violet values refer to a better / worse representation of the data by the reference (seasonal-only model). The reference weighting of other non-exceedance probabilities are barely different. Reference weighting plotted against station altitude (b, black dots) indicates better performance for the reference in the lowlands than in mountainous regions. The subset QSS (second axis, blue crosses) of the seasonal-interannual model do not show an improved skill for stations at higher altitude.

Besides the monthly contribution to the station-wise skill for *Wesertal-Lippoldsberg* Fig. 10 shows as well the results for the other three example stations. *Rain am Lech* serves as an example with a very high skill mainly dominated by a better reflection of the data for the months May and September. At *Mühlhausen / Oberpfalz-Weihersdorf* the return levels for spring can be estimated slightly better than for autumn and *Krümmel* is dominated by a positive skill for June and July.

In summary, it can be noted that modelling interannual variations are beneficial for estimating return levels for all months, especially for the summer season. However, at a few stations the flexible modelling leads to a partly worse representation, in particular for larger return periods. Both, seasonal modelling and seasonal-interannual modelling may have difficulties to capture mechanisms for precipitation formation in alpine regions.

## 6  Importance of a flexible shape parameter

Analysing the selected models of the 519 considered stations shows that about 34 % (178 / 519 stations) prefer a model with interannual and/or seasonal variations in $\xi$. Fig. 11 illustrates the spatial occurrence and the kind of variation (seasonal, interannual or interaction). It is noticeable that the density of stations with a flexible $\xi$ is much higher in the north and east of Germany than in the south. The reason for a located-dependant variable shape parameter is an interesting question for further studies. We assume that different meteorological processes play a major role, e.g. the influence of weather types or the kind of precipitation (stratiform or convective). A dependence of flexibility in $\xi$ on the record length is not obvious (not shown). Most of the stations (106 / 178, about 60 %) are represented by a model including seasonal variations, whereby many of them (92 /106 stations) do not favor an interannnually varying shape parameter at all. Only a few stations (17 / 178, about 10 %) prefer a model with direct interannual changes. Nevertheless, two regions with a slight agglomeration of direct interannual changes can be detected: in the middle of *Bavaria* (Federal State in the south-east) and in the northeast around the *Mecklenburger Seenplatte* represented by the example station *Krümmel*. About 39 % of the stations (69 / 178) show a interannually varying seasonal cycle (interactions); these stations are almost uniformly distributed across Germany with a somewhat higher density around the example station *Wesertal-Lippoldsberg*.

We a) quantify the gain from a flexible shape parameter with respect to a model with constant $\xi$ and b) analyse the contributions of the seasonal, interannual and interacting variations. To this end we use four model selection setups with focus on $\xi$: setup 1) with constant $\xi$, setup 2) with seasonal components in $\xi$, setup 3) with seasonal and interannual components in $\xi$, setup 4) with seasonal and interannual components, as well as their interactions in $\xi$. All other parameters are allowed to have seasonal, interannual and interacting components in all setups. Fig. 12 illustrates the gain in performance for the different steps. The monthly skill for seasonal-interannual variations including interactions against a constant $\xi$ averaged over the 178 stations expressed as the contribution to the total QSS is depicted in the top panel. There is positive skill for all months and return periods (with some exceptions with slightly negative values). The highest contribution to the total skill arises from the summer months, for which the reference weighting is increased (Sec. 5).

To analyse the contribution of the seasonal component to the skill, we use setup 2 (seasonal-only in $\xi$) against a constant $\xi$ (setup 1). Setup 2 results in a variable shape parameter for 106 of 178 stations; for the rest of the records a model with

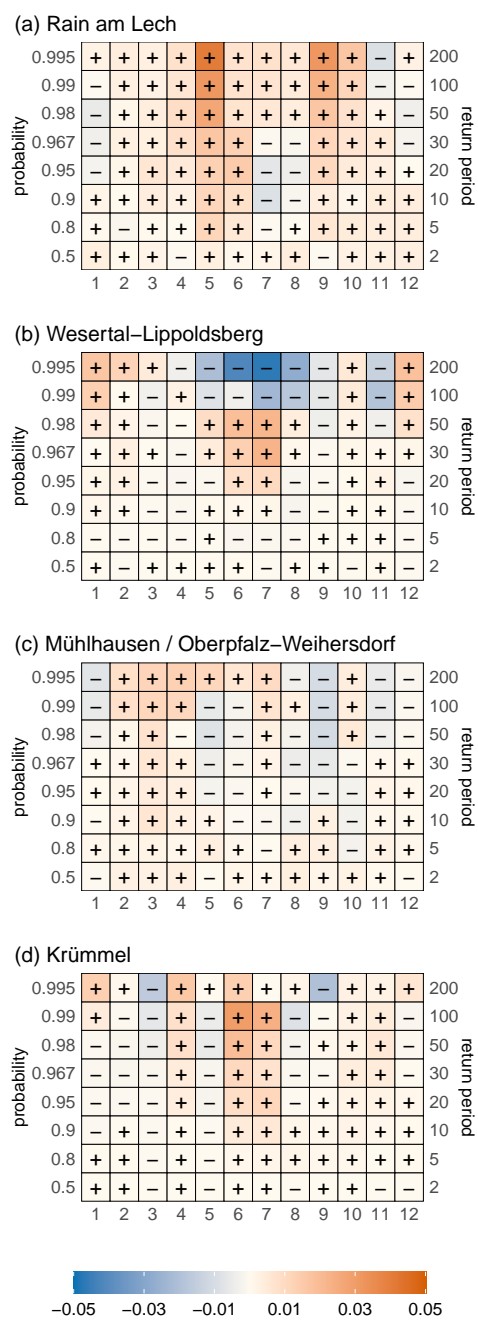

**Figure 10.** Monthly contribution (x-axis) to the station-wise QSS depicted in Fig. 8 a) for the example stations shown for different non-exceedance probabilities (left y-axis) / return levels (right y-axis). Positive / negative values (orange / blue) indicate a gain / loss in skill of the seasonal-temporal model with respect to the only seasonal model.

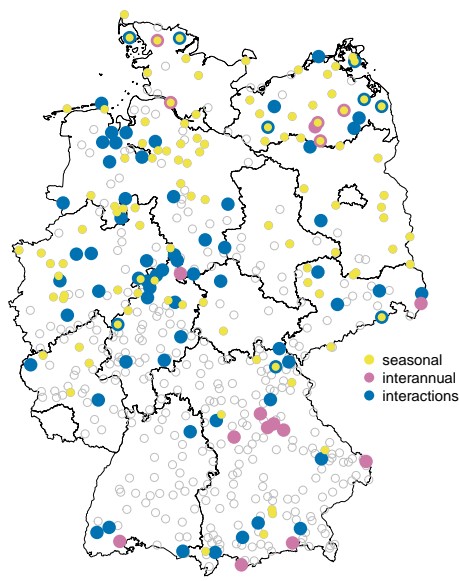

**Figure 11.** Spatial distribution of stations with flexible shape parameter $\xi$. Seasonal variations (yellow) occur at 106 / 178 stations, interannual (pink) at 17 / 178 and interactions (blue) at 69 / 178 stations. Grey circles mark the stations without variations in $\xi$.

no variations in $\xi$ was prefered. The skill of these 106 stations with respect to setup 1) is depicted in the left panel of Fig. 12. Seasonal flexibility in $\xi$ improves in particular the return levels for summer; there is a very small gain in winter. For the transitional months March/April and September the increased flexibility led to a slightly worse model. A change in the

355 shape parameter could indicate a change of the dominating precipitation type (convective in summer, stratiform in winter). The flexible modelling do not benefit for months characterised by the transition of the precipitation regime, since no dominating precipitation type exists. A variable $\xi$ for the season mainly improves the return levels of the higher return periods, while the skill for two and five years are slightly decreased.

  Setup 3 evaluates the gain by adding an interannual component to $\xi$ with respect to a seasonal-only model (setup 2). Only 17

stations prefered this type of model but for those, we found an improvement of the return level estimates for all months (bottom panel). Only for the transitional months March and September the interannual variations do not improve the performance. Additionally, the lower return periods of two and five years do not benefit from this flexibility.

  In setup 4 we allow additionally for interactions and compare the selected models with those chosen in setup 3 (seasonal-interannual model without interactions). The skill averaged over 69 stations with interaction terms for $\xi$ is shown in the right

plot of Fig. 12. The interactions improve the return levels for all months and return periods, especially for the summer months, and are able to faintly compensate the lack of skill for the lower return periods and the transitional months. The skill shown in Fig. 12 is averaged over the respective stations, however the performance for the individual stations can differ. For example, as already mentioned in Sec. 5 and analysed in more detail in Sec. 7, the 100-year and 200-year return levels at the example station

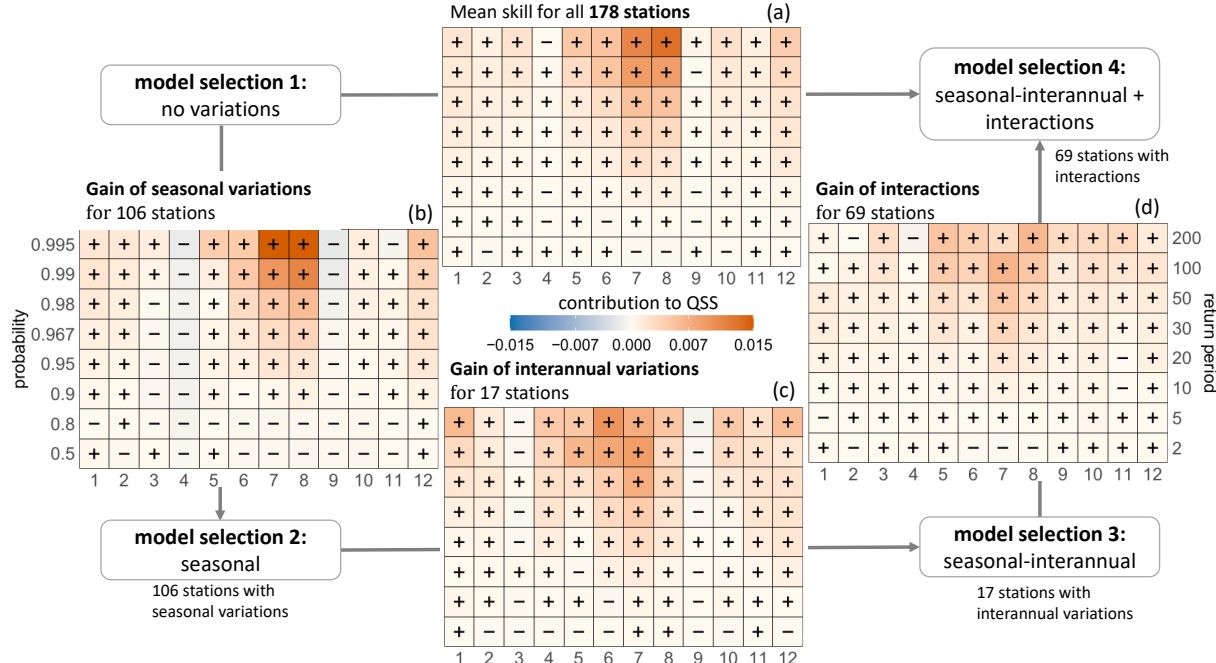

**Figure 12.** Scheme for analysing the importance of and performance gain by a flexible shape parameter as contribution to the total QSS. Illustrations (axes, colors, signs) equal to Fig. 10. The gain of adding seasonal variations in $\xi$ (left plot) is analysed for 106 stations as a result from a model selection with only seasonal components in $\xi$ (setup 2). A model selection with constant $\xi$ (setup 1) is used as reference. The bottom panel shows the gain of seasonal and interannual components (17 stations, setup 3) with respect to a seasonal-only $\xi$ (setup 2). The right panel finally show the gain by allowing interactions (setup 4) with repect to setup 3 (without interactions) for 69 stations. The skill of a flexible shape parameter (seasonal, interannual and interactions) with respect to a constant $\xi$ is shown in the top panel for 178 stations.

*Wesertal-Lippoldsberg* are overestimated (visually obtained by comparing return levels and observations) for the last decades
resulting in a worse representation of the most recent data. The overestimated return levels can be explained by considering
the seasonal cycle of the shape parameter $\xi$ for different years (1931, 1976, 2021) in Fig. 13. As depicted in Fig. 5 the shape
parameter at this station can be expressed with $\xi = \xi_0 + \xi_{1,1,\cos}^+$ (according to Eq. 13), i.e. a linear rise in the amplitude. Thus,
for the first observational year 1931 the amplitude is modelled to be zero and increases linearly with time reaching its maximum
for the last year in the record (2021). While for earlier years this linear change represents quite well the extreme precipitation,
for the end of the record the values of $\xi$ especially for summer become very large, which can not be supported by the sparse
database.

Additionally, Fig. 13 shows the seasonal cycle in $\xi$ for the example station *Krümmel* whose shape parameter is composed to
$\xi = \xi_0 + \xi_{1_{\cos}} + \xi_{3_P}$. Seasonality remains unchanged while the direct effect of the third Legendre polynomial leads to a shift of
the cycle to smaller / larger values of $\xi$ in 1976 / 2021 compared to 1931, leading to pronounced variability in the return levels
of this station. The seasonal cycles for the example stations will be disscused in more detail in Sec. 7.

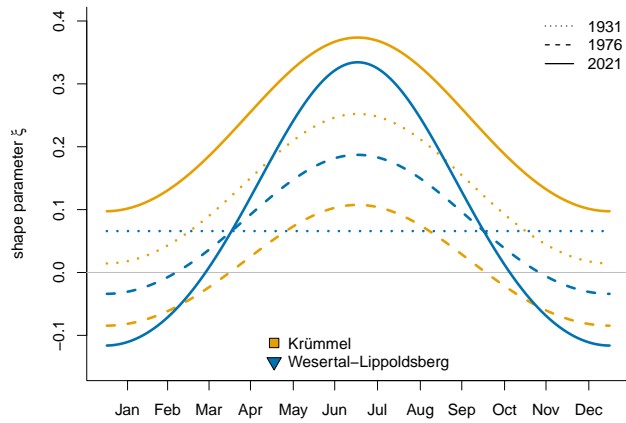

**Figure 13.** Seasonal cycle of the shape parameter $\xi$ for the example stations *Krümmel* (orange) and *Wesertal-Lippoldsberg* (blue) for the years 1931 (dotted), 1976 (dashed) and 2021 (solid). The station symbols in the legend are selected according to the stations position of Fig. 1.

A negative shape parameter is unusual for describing the GEV distribution of extreme precipitation (Papalexiou and Koutsoyiannis (2013), Ragulina and Reitan (2017)) since the resulting distribution is characterised by an unnatural upper bound. In our analysis the shape parameter is able to change with time such that negative values for $\xi$ for a certain period are considered to be unproblematic.

In general a varying shape parameter leads to a better representation of the data for all months and return periods in particular for the very extreme events in summer; only for very few stations the flexibility leads to a worse skill of return level estimates.

## 7   Impact of climate change on the seasonality of extreme precipitation

In this Section we aim to assess the impact of climate change on seasonal extreme precipitation (RQ3). With a simple linear model for each month and station we quantify the interannual variation of return levels for a given non-exceedance probability.
We compare the time period from 1941 to 2021 where all stations have data. Note that estimating linear trends for fixed (and short) periods of time can yield very different results depending on the considered time period due to decadal variability. Thus, the trend estimates presented here for the given time period serve as a rough indicator for climate change effects; for a more detailed analysis the whole datasets should be taken into account for each station. Appendix A explains the calculation of the linear trend in more detail. Fig. 14 illustrates the proportion of stations with a positive, negative or no trend for (a) the 2-year,
(b) 10-year, and (c) 100-year return levels. The trends are stated in relative changes from 1941 to 2021. Changes are mainly very weak ($< 5\%$); only 15% to 35% (depending on the month and occurrence probability) show more pronounced trends. In general an increase of the return levels occur more often than a decrease for all return periods, especially in June (more than three times more often). Only for the return levels in April a decline slightly prevails. The patterns of the 5- to 200-year return

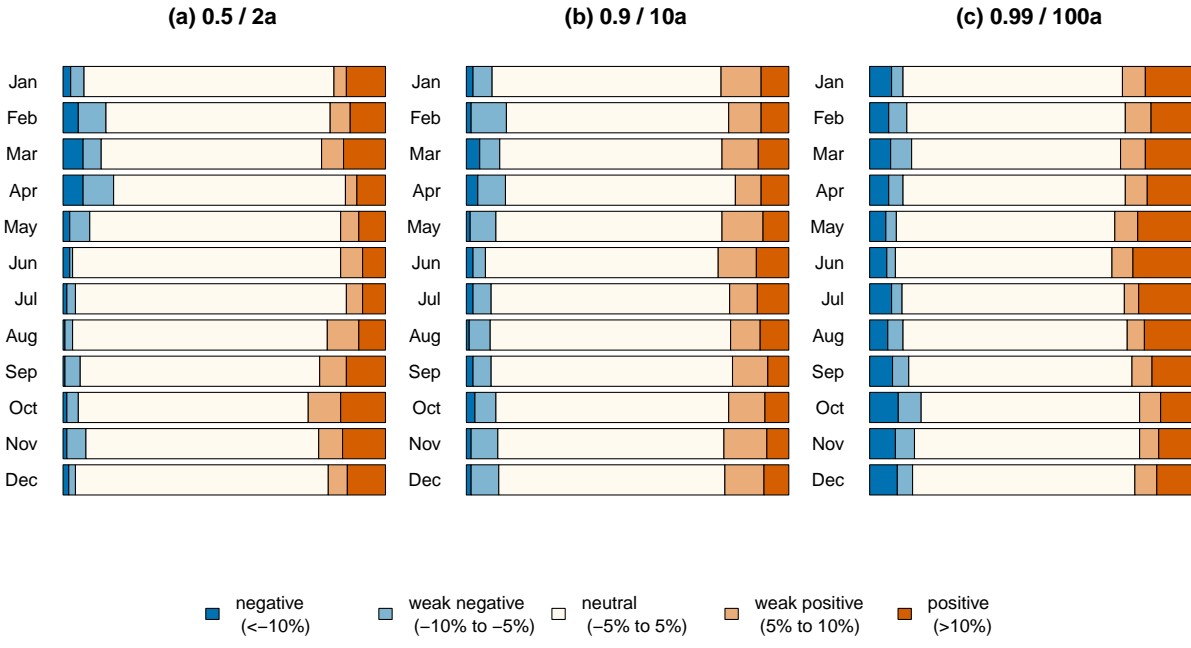

**Figure 14.** Proportion of stations with a positive (light/dark orange), negative (light/dark blue) or neutral (white) relative change from 1941 to 2021 for (a) 2-year return level (p=0.5), (b) 10-year return level (p=0.9) and (c) 100-year return level (p=0.99) for the months January to December (rows).

levels are similar but with smaller trends for the shorter return periods. The characiceristics of the 2-year return level differ: an
400 increase is more often visible for the months March and September to November with a less pronounced signal for the summer months. In contrast to that, the trends of the 100-year return level are stronger in summer.

About 35% to 50% of the considered 338 stations (121/338 2-year return level, 164/338 10-year return level, 170/338 100-year return level) show a change larger than 5% for at least one month of the year. The trends are regionally very different (maps for 2-year and 100-year return levels are given in Appendix B). Despite them partly very small-scaled characteristics,
uniform behaviour for several regions can be detected. Two of these regions with more pronounced changes are considered in more detail: one in southern Germany represented by the station *Rain am Lech* and the other one in the center of Germany exemplified by the station *Wesertal-Lippoldsberg*.

The 2-, 10-, and 100-year return levels for the station *Rain am Lech* are depicted in Fig. 15 (a). Besides the interannual changes (a.1), the seasonal cycle for the first/last record year (1899/2021) and the first year of the common time period (1941)
are shown (a.2). The return levels of this region are characterised by an increase for all months and return periods. For instance, the largest 100-year return level in the year occuring in summer rose from 54.6 mm/day in 1899 to 86.0 mm/day in 2021, corresponding to an increase of about 58%. Considering the 100-year return levels of the seasonal-only model demonstrates that

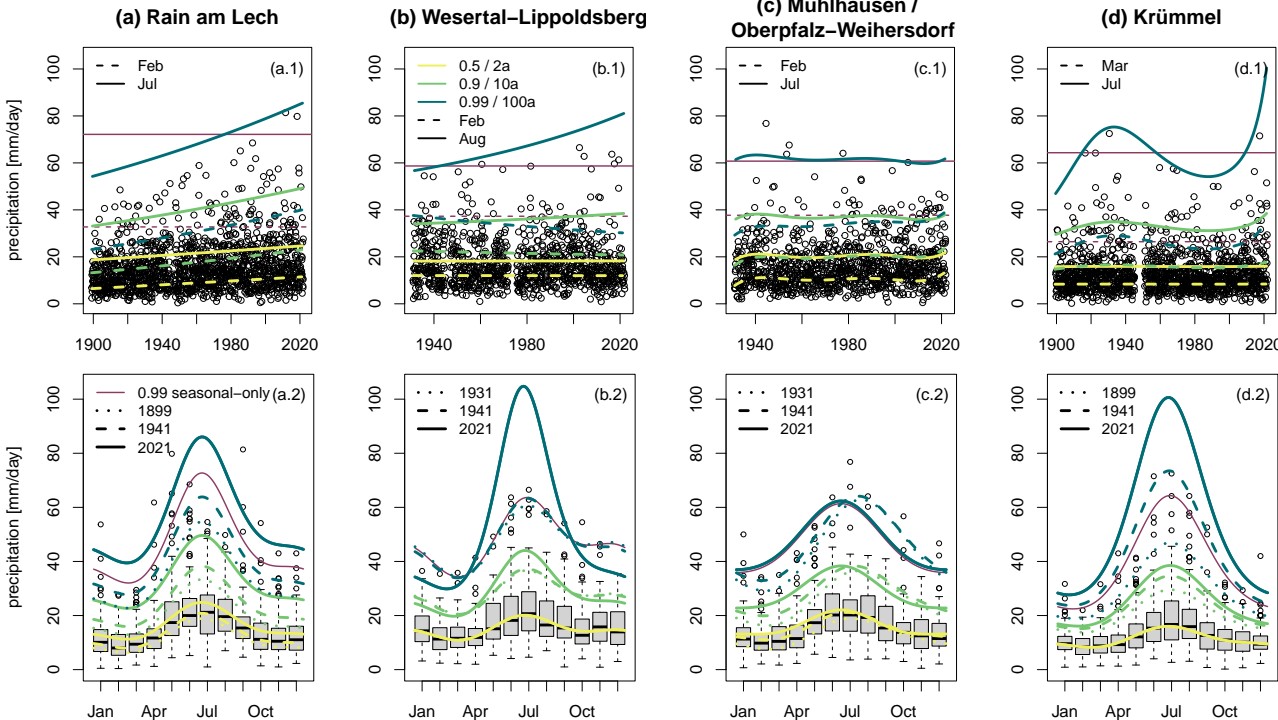

**Figure 15.** Observations and return levels for the stations *Rain am Lech* (1899-01-01 until 2021-12-31) (a), *Wesertal-Lippoldsberg* (1931-01-01 until 2021-12-31) (b), *Mühlhausen / Oberpfalz-Weihersdorf* (1931-01-01 until 2021-12-31) (c) and *Krümmel* (1899-01-01 until 2021-12-31) (d). The top row shows the observations (dots), the 2-year (yellow), 10-year (green) and 100-year return levels (blue) for the months with lowest / highest return level (dashed / solid) against time (years, x-axis). Additionally, the 100-year return levels of the seasonal-only model are depicted for the same two months (burgundy). The bottom row depicts the seasonal cycle of the return levels for the first observation year (dotted), 1941 (dashed) and 2021 (solid) and the observations as Box-Whisker-Plots. Additionally, the 100-year return levels of the seasonal-only model (burgundy) are depicted in both rows as well.

a non-interannual approach leads to highly underestimated values especially for the first record decades. The model selection reveals that not only the location parameter changes linearly with the years but as well the scale parameter (Fig. 5). The model
verification (Fig. 10) confirms, that the trend in the return levels is necessary for an adequate describtion of the observations especially for the transitional months of May, September and October. Thus, an increase of extreme precipitation amounts as expected from the antropogenic climate change can be seen very clearly for this region.

The second region, which is considered in more detail, is characterised by a decrease of return levels in winter and an increase in summer leading to a rise of the seasonal cycle's amplitude. Fig. 15 (b) shows the 2-, 10-, and 100-year return levels for the
station *Wesertal-Lippoldsberg*. Since the interannual change for this station is best described by a model with a flexible shape parameter only (Fig. 5), the 2-year return levels remain constant with the years. Towards higher return periods, changes are more prominent. They are pronounced for summer and winter, while the transitional months March/April and September/October

remain unaltered. The change of the seasonal cycle could be attributed to a combination of different processes. On the one hand a higher water content of the air due to a temperature rise leads to an increased potential for extreme precipitation, particularly pronounced in summer. On the other hand climate change can affect the characteristics of weather types and large-scale atmospheric circulations (e.g. NAO), which could result in a change of extreme precipitation as well in winter. The model verification (Fig. 10) confirms that a model with a changing seasonal cycle better represents the data observed in summer for return periods of 10 to 50 years, while the 100- and 200-year return levels are strongly overestimated with respect to the observations, especially for the most recent decades. In constrast, the seasonal-interannual model is more beneficial for estimating winterly return levels with return periods longer than 30 years. These characteristics can be seen as well by comparing the 100-year return levels of the seasonal-interannual model with those of the seasonal-only model.

In addition to a change of the precipitation's magnitude, a phase shift can influence the risk of damage as well. Therefore, we analyse as well the linear change in the phase expressed as the day in the year with the highest return level for the time period 1941-2021. Here, a simple linear model is adequate for the cyclic variable since a shift of the day with the highest precipitation from December to January or vise versa do not happen at all. The change of the phase in days for different return periods is illustrated in Fig. 16 (a). More than two-third of the stations show less pronounced changes (< 5 days). In general a shift to earlier times in the year appears more frequently for almost all return periods except of the 2-year return level. For the latter, several different regions with strong shifts to later and only one large contiguous area in the north with a shift to earlier times can be detected (b). For the 100-year return level (c), such distinct regions do not appear, but in gerneral a shift to earlier times prevails for the whole country. The latter behaviour can be appreciated in station *Mühlhausen / Oberpfalz-Weihersdorf*, whose 2-, 10-, and 100-year return levels are depicted in Fig. 15 (c). The shift of the seasonal cycle to earlier times leads to increased return levels for the first half of the year, such that the 100-year return level in spring is about 13 mm/day higher in 2021 than it was in 1931. The decrease in autumn appears less strong with a maximum change of the 100-year return level of about -8 mm/day. During the 90 years of observations the annual maxima of the 100-year return level has shifted forward by 35 days. Since only a shift and not a rise of the seasonal cycle occurs, the analysis of annual maxima would not show any changes. In gerneral, a shift of the seasonal cycle to earlier times lead to an increased risk potential. The probability of flooding events rises since snow melting and heavy precipitation coincide in spring. Additionally, higher crop losses may occur since plants are more vulnerable to extreme precipitation during early growing stages. Although differences between the 100-year return levels of the seasonal-only and the seasonal-interannual model are not very pronounced, the shift from late summer to early summer, which might be continued in future, can not be detected with the non-interannual approach.

The example station *Krümmel* (Fig. 15) (d) shows neither a linear change of the return levels nor a phase shift but points out other interesting features. It serves as a representative of the region *Mecklenburger Seenplatte*, since several neighboring records show similar characteristics. Here, pronounced climate variability can be detected in the seasonal cycle of extreme precipitation which might be important for risk assessment and the design of hydraulic structures. Due to those climate variability it can be illustrated that the commonly used stationary approach for a fixed historical time period can lead to erroneous return levels. For example, in Germany the stationary return levels based on the observations since 1951 have been using for infrastructure planning (DWD (2000); KOS (2022)). That means for the example station *Krümmel* the heavy precipitation events from the

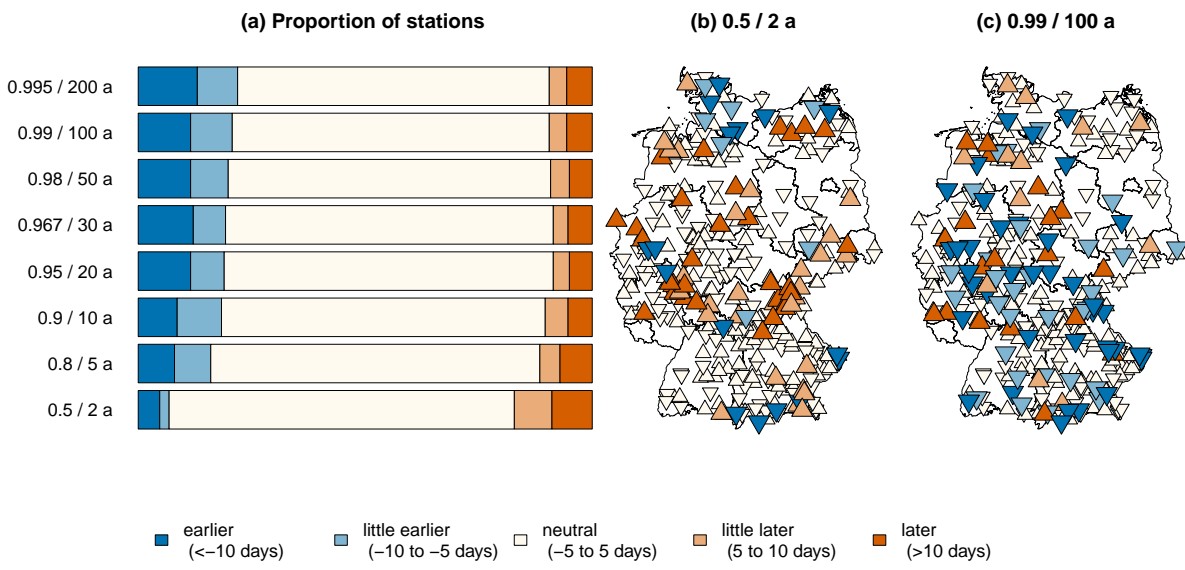

**Figure 16.** Phase shift in days from 1941 to 2021 for the non-exceedance probability / return period of (a) 0.5 / 2 years, (b) 0.8 / 5 years, (c) 0.9 / 10 years and (d) 0.99 / 100 years. A shift to later / earlier times are marked with pointing up triangles / pointing down triangles. Minor changes (< 5 days) remain uncolored, stronger shifts to later /earlier times are highlighted in orange / blue.

1930s have been discarded, potentially leading to underestimated return levels and to too small dimensioned hydraulic systems for the precipitation of recent years. A non-stationary approach including the whole dataset can improve the accuracy of the
return levels. For this example, the seasonal-only approach applied to the whole record might be beneficial in terms of long-term risk assessment and hydraulic design since natural variability does not play a key role for longer planning horizons. However, for short- to mid-term risk assessment, e.g. for agriculture or tourism sector, the natural variability might be of relevance.

We sum up that monotonous trends are spatially different and mainly weak compared to return level uncertainties (not shown). Nevertheless, we detect regions with common and more pronounced changes. In general, the characteristics of the
2-year return levels differs from those of longer return periods.

## 8   Conclusions

### 8.1   Summary

We analyse seasonal-interannual variations of extreme precipitation at 519 stations (with at least 80 years of observations until 2021-12-31) in Germany using a non-stationary block maxima approach. The three parameters of the Generalized Ex-
treme Value distribution (GEV) are allowed to vary with the months (seasonal variation) and the years (interannual variation), whereby the seasonal variations are captured with a series of harmonic functions and the interannual variations with Legen-

dre Polynomials with a maximum power of five. Interactions between seasonal terms (months) and interannual terms (years) allow the description of a interannually varying seasonal cycle. Since we consider higher polynomial orders than linear, the models are able to reflect other than linear trends, e.g. more complex climate variability. A step-wise model selection based on the Bayesian Information Criterion (BIC) identifies a suitable model for each station separately which is used to calculate seasonal-interannual changing return levels for different return periods (non-exceedance probabilities). To validate the models we use a leave-one-year-out cross-validated quantile score to measure the model performace for individual quantiles (return-levels). The Quantile Skill Score (QSS) and its decomposition for stratified verification provides additional information about the skill of the model with respect to a only seasonal varying non-stationary GEV. We addressed three research questions:

### 8.1.1   RQ1: Can a model with interannual variations better represent the observations than a seasonal-only model?

For 334 / 519 stations (about 65%) the BIC favours a model with interannually varying return levels. For the other stations, the BIC based model selection strategy do not give any evidence for a model more complex than the one with only seasonal variation. For the 334 selected records, the cross-validated verification confirms that the models with interannual variations yield a more adequate describtion of the data than models with only seasonal variations. Only for very few stations the seasonal-interannual return levels are more inaccurate in particular for higher return periods in summer. A stratified verification along months ascertains that modelling interannual variations are more beneficial for summerly extreme precipitation. A stratified verification along different stations points out a lack of capturing important mechanismn for extreme precipitation in mountainous regions by only modelling seasonal and interannual variations.

### 8.1.2   RQ2: How important is a flexible shape parameter to reflect recorded variations?

As the shape parameter $\xi$ describes the behaviour of the most sparse events, it is considered to be difficult to estimate and hence frequently kept at a constant but estimated value in other works. The BIC based model selection strategy favours a flexible shape for 178 / 519 stations (about 34%), whereby about 52% (92/178) of these records prefer a seasonal-only component. For the remaining stations with variable $\xi$, an interannually changing seasonality occurs more often than the direct interannual variations. Furthermore, we find many of the records with $\xi$ varying with season and/or years in the north and east of Germany. We suggest that location-related weather regimes might be responsible. The spatial distribution of stations described with an interannually varying shape parameter provides an interesting topic for further investigations. In our study the flexible shape parameter leads to a better representation of the observations for all months and return periods; this is particularly evident for the very extreme events in summer. A stepwise addition of seasonal, interannual and interactional variations in $\xi$ enables an analysis of the performance of those individual components. All three components lead to improved return levels. The seasonal and interannual variations mainly improve the statistical models' representation of the summerly and winterly return levels with longer return periods (20yrs to 200yrs) while interactional variations are favourable for all months and return periods.

### 8.1.3 RQ3: How does climate change affect the seasonal cycle of extreme precipitation in Germany?

To quantify the consequences of climate change for the seasonal cycle, we obtain linear trends of the interannually varying return levels and the phase of the seasonal cycle (day in year with the highest return level) for the common analysis period from 1941 to 2021. A unambiguous signal in these trends which could be related to climate change can not be found since only about one fifth to one third of the 519 considered stations (2-year return level: 23%; 10-year return level: 32%; 100-year return level: 33%) show a stronger linear change (>5%), either positive or negative, for at least one month of the year. However, in general an increase of the return level is more prevalent than a decrease. The 2-year return levels mainly rise during spring and autumn, while for the 100-year return period the trends are more pronounced in summer. Nevertheless, for many of the records the trends of the return levels are weak. Trends are regionally very different; for some areas the changes are more pronounced. We assume spatially independent datasets, although we cannot exclude that the same large-scale precipitation event causes similar trends for neighboring stations. By means of example stations, two regions with different changes of the seasonal cycle are considered in more detail. In parts of southern Germany extreme precipitation is characterised by rising return levels for all months of the year with 50% higher values in 2021 than for the beginning of the 20th century for some stations. In parts of the center of Germany the amplitude of the seasonal cycle increases due to higher/lower return levels in summer/winter. The phase shift of the seasonal cycle regionally diverge but in general extreme precipitation occurs earlier nowadays. Depending on the timing of snowmelt, this may lead to a higher risk potential due to the coincidence of heavy precipitation and melting snow masses but as well for crop losses since plants are more vulnerable to extreme precipitation in earlier growing stages. Only for the 2-year return period several different distinct regions show a shift towards the later year.

### 8.2 Discussion

Since extreme precipitation is highly variable in time and space and long datasets are rare, coherent outcomes of different research studies are crucial for a suitable risk assessment and risk adaptation. Zolina et al. (2008) and Łupikasza (2017) analysed the seasonal 0.95 and 0.99 quantile of daily precipitation sums using quantile regression and detected an increase in spring, autumn and winter for the period of 1950-2004 and 1950-2008 in Germany, while summer quantiles decrease. Their results seem to be in contradiction to our findings of more intense heavy precipitation in summer. These differences could have various reasons. First of all, the time period considered for the linear trends are different which could be decisive in particular if pronounced climate variability exist. Furthermore, we consider a more recent dataset (17 and 13 more recent years). An investigation on the damage related to extreme precipitation in Germany indicates intensified heavy precipitation events during the last decade (Trenczek et al., 2022). Finally, the methods introduced in this paper and those of the references mentioned above are different. While our analysis is based on extreme value statistics for block-maxima, Zolina et al. (2008) and Łupikasza (2017) consider precipitation sums for all days; both have different interpretation of resulting quantile information. Furthermore, Zeder and Fischer (2020) detected a positive connection between extreme precipitation over Germany and the rising north-hemispheric temperature as well for summer.

The pronounced climate variability in extreme precipitation which can be detected at the example station *Krümmel* partly
fits to the results of Willems (2013) discovering multidecadal oscillations with more often and intense extreme precipitation
events in northwestern Europe in the 1910s, 1950-1960 and since 2000, while in south-western Europe the oscillation is anti-
correlated with highs in the 1930-1940s and 1970s. Willems attributed the multidecadal variations to oscillations in North
Atlantic climate and determined a coincidence of pressure anomalies between the Azores and Scandinavia (ASO index) and
extreme precipitation in winter. Periods with increased summer extreme precipitation are explained with the occurrence of more
cyclonic weather types. Unfortunately, station *Krümmel* did not record for a few years around 1950 and thus the interannual
variability around this time cannot be verified due to the data gap. The variations at the station *Krümmel* roughly fits to the
North Atlantic Oscillation (NAO) as well (Hurrell (1995); Hurrell and Deser (2010)), at least for the winter months, although
Willems (2013) reported a weaker relation between NAO and extreme precipitation.

An understanding of physical mechanisms leading to the observed results was not in the focus of this study but needs to
follow. We imagine a combination of increased convection due to higher surface temperatures and moisture (Westra et al.,
2014; Aleshina et al., 2021), as well as changes of large-scale atmospheric circulations (Casanueva et al., 2014).

Seasonal and interannual variation in extreme precipitation can be described with a combination of harmonic functions and
orthogonal polynomials like the Legendre polynomials. For this investigation but as well for previous studies, the latter one
has proven to be helpful to approximate highly non-linear variations. However, their nature having the highest/lowest values at
the borders of the time period potentially lead to very high or low return levels for the beginning and the end of the time series.
This could mislead the analysis of trends. A possible strategy to prevent the boundary problem is to select a slightly larger
scaling area than the period observed for obtaining the Legendre polynomials.

A possible application of the presented seasonal-interannual approach in the field of risk adaptation could be realised by
calculating design-life levels. This concept has been introduced by Rootzén and Katz (2013) and widely applied in research
and risk management (e.g. Thomson et al., 2015; Mondal and Daniel, 2019; Xu et al., 2019; Byun and Hamlet, 2020). The
design-life level is a measure for quantifying and communicating environmental risks in a changing climate accounting for the
service life of a system (design-life period, e.g. 30 years) and the time, when the system will be installed (e.g. in 2025). Due to
changing extreme precipitation characteristics, the 2025-2055 1% design-life level could be different from the 2055-2085 1%
design-life level. More detailed explanations and example calculations can be found in Appendix C. The seasonal-interannual
modelling approach can be used to calculate future seasonal design-life levels either by extrapolating past climate trends or by
applying to outputs from climate projections. Since for risk adaptation in an engineering context annual design-life levels are
more beneficial then seasonal ones, the same methodological concept can be applied to obtain annual values out of a seasonal
modelling approach (Maraun et al., 2009; Fischer et al., 2018).

## 8.3 Outlook

Extreme precipitation is influenced by many different effects (e.g. location, air-temperature, large-scale atmospheric circula-
tion, lifting effects, ...), most of them are highly non-linear and difficult to quantify in terms of their role. In this study, we
utilize the time as a covariate since it can be seen as a proxy combining those different unknown effects. Based on our results,

the consequences of climate change could be assessed in more detail by using surface temperature, greenhouse gas emissions or indices of large-scale atmospheric circulations patterns as terms in the predictor. This offers also an opportunity to evaluate the climate variability of extreme precipitation and the processes associated with it.

As above discussed, the interannual variability of one example stations visually matches the results of Willems (2013) and the North Atlantic Oscillation (NAO) (Hurrell (1995); Hurrell and Deser (2010)). For robust conclusions in this respect, our findings might be used as a starting point for a more detailled analysis. Determining the responsible mechanisms for the climate variability of seasonal extreme precipitation will not only enhance the understanding of the connections but also will improve the heavy precipitation datasets of climate models, since the predictability of those mechanisms (e.g. NAO, surface temperature) are often better than for precipitation.

Additionally, trends might differ for different durations of the precipitation events, changes for e.g. hourly or sub-hourly extreme precipitation are worthwhile to consider apart from daily precipitation sums. Typically, observation records of higher resolved extreme precipitation are shorter and hence analysis of interannual variability is more uncertain. One possibility to improve accuracy is to use a smooth relationship between different durations directly in the formulation of the GEV (e.g., Ulrich et al., 2020, 2021) for an effective data usage by considering different durations simultanously.

Furthermore, a different approach for modelling the interannual variations could be considered: to overcome the boundary problem of the Legendre polynomials it might be worthwhile to consider different orthogonal polynomials, e.g. the first kind of the Chebyshev Polynomials, or to use a vector generalized additive model (VGAM, Yee (2015b)) to become smooth, non-parametic variations. An extrapolation of the calculated values towards the design-life period (e.g. for the next 50 years) is required and should be carried out carefully. The corresponding design-life levels (Rootzén and Katz (2013)) form the basis for the construction of the hydraulic systems. However, this requires a modelling strategy being able to reliably estimate future return levels.

In our investigation we consider return level estimates. However, analysing their uncertainties are crucial. For further investigations, confidence intervals, e.g. calculated with the delta method (Coles, 2001), should be taken into account. A comparison of uncertainties evolved by the seasonal-interannual model and those of a seasonal-only model could deepen the investigation if interannual models are beneficial for risk assessment or if the changing return levels are rather within the uncertainty range of non-interannually varying return levels.

## 8.4   Main Achievments

We introduce a seasonal-interannual modelling approach to assess variations of extreme precipitation leading to more accurate return levels. The interactional consideration enables a modelling of a changing seasonal cycle in form of a changing amplitude and / or phase. The approach is able to reflect long-term changes and climate variability. In addition, we show that a flexible shape parameter of the GEV is beneficial. Finally, we use the approach to detect regions in Germany for which extreme precipitation is likely to be affected by climate change. In general, changes are weak, however, an increase is prevalant compared to a decrease. The lower extreme precipitation rises generally in spring and autumn and its seasonal cycle is shifted to later times in the year, heavy precipitation increases mainly in summer and occurs earlier in the year.

## Appendix A: Linear Trends in Return Levels and Phase

The linear trend in return levels and phase of seasonal cycle is calculated for each stations, months and occurrence probability separately using a simple linear model. The relative change from the first to the last year included in the linear model is obtained with:

$$c_\% = \frac{v_l - v_f}{v_f} \cdot 100\% \tag{A1}$$

with $c_\%$ being the relative change and $v_f$ / $v_l$ the first and the last value of the linear regression line. Fig. A1 illustrates exemplarily the dependence of the selected time period on the linear trend. While the relative change in the 100-year return level at the station *Krümmel* for the period 1899-2021 equals to 3.17%, the return level in 2021 is increased by 8.16% with respect to 1941. According to the rating scheme of Fig. 14 the first belongs to a neutral and the latter to a weak positive trend. Thus, linear trends for fixed (and short) time periods should be regarded with care.

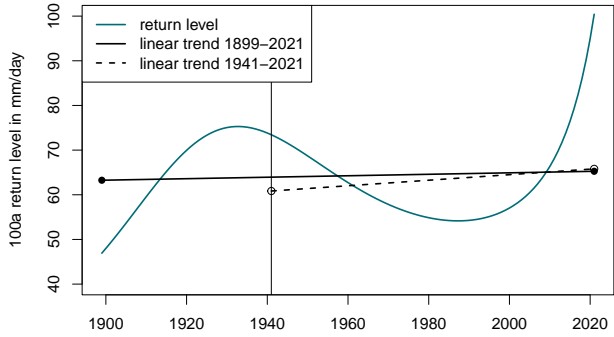

**Figure A1.** 100-year return level in mm/day for station *Krümmel* (blue), the linear trend for the whole time period (black, solid) and the linear trend for the period 1941-2021 (black, dashed). Dots mark the first and the last value of the respective regression line.

## Appendix B: Maps of relative changes in return levels

Fig. B1 and Fig. B2 show the relative changes in the 2-year return level ($p = 0.5$) and the 100-year return level ($p = 0.99$) for the 338 stations with interannual variations. Changes are regionally divergent, however, several contiguous regions are visible.

## Appendix C: Design-life level

According to Rootzén and Katz (2013), the design-life level is a measure to quantify risks for engeneering design purposes in a changing climate. This measure can be regarded as a logical extention of the return level approach which can only be meaningfully interpreted in a stationary setting. For example, a 100-year return level of extreme precipitation is the value which is expected to be exceeded in mean once in hundred years. Due to changing climate, an event can occur in 2023

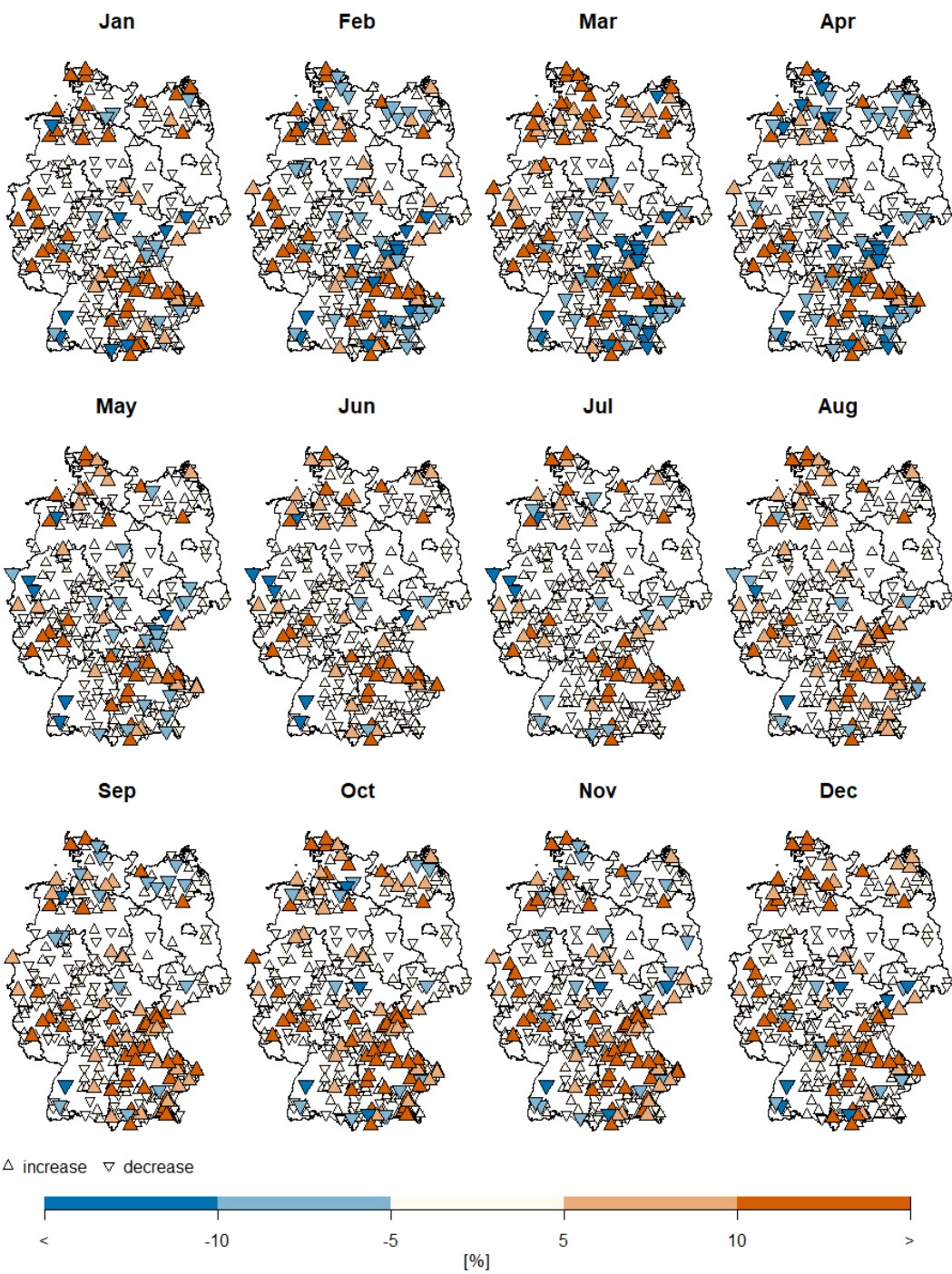

**Figure B1.** Relative change from 1941 to 2021 for the 2-year return level (non-exceedance probability of $p = 0.5$) for 338 stations. Increases / decreases are marked with pointing up triangles / pointing down triangles. Minor changes (< 5 %) remain uncolored with small symbols, stronger increases / decreases are highlighted in orange / blue.

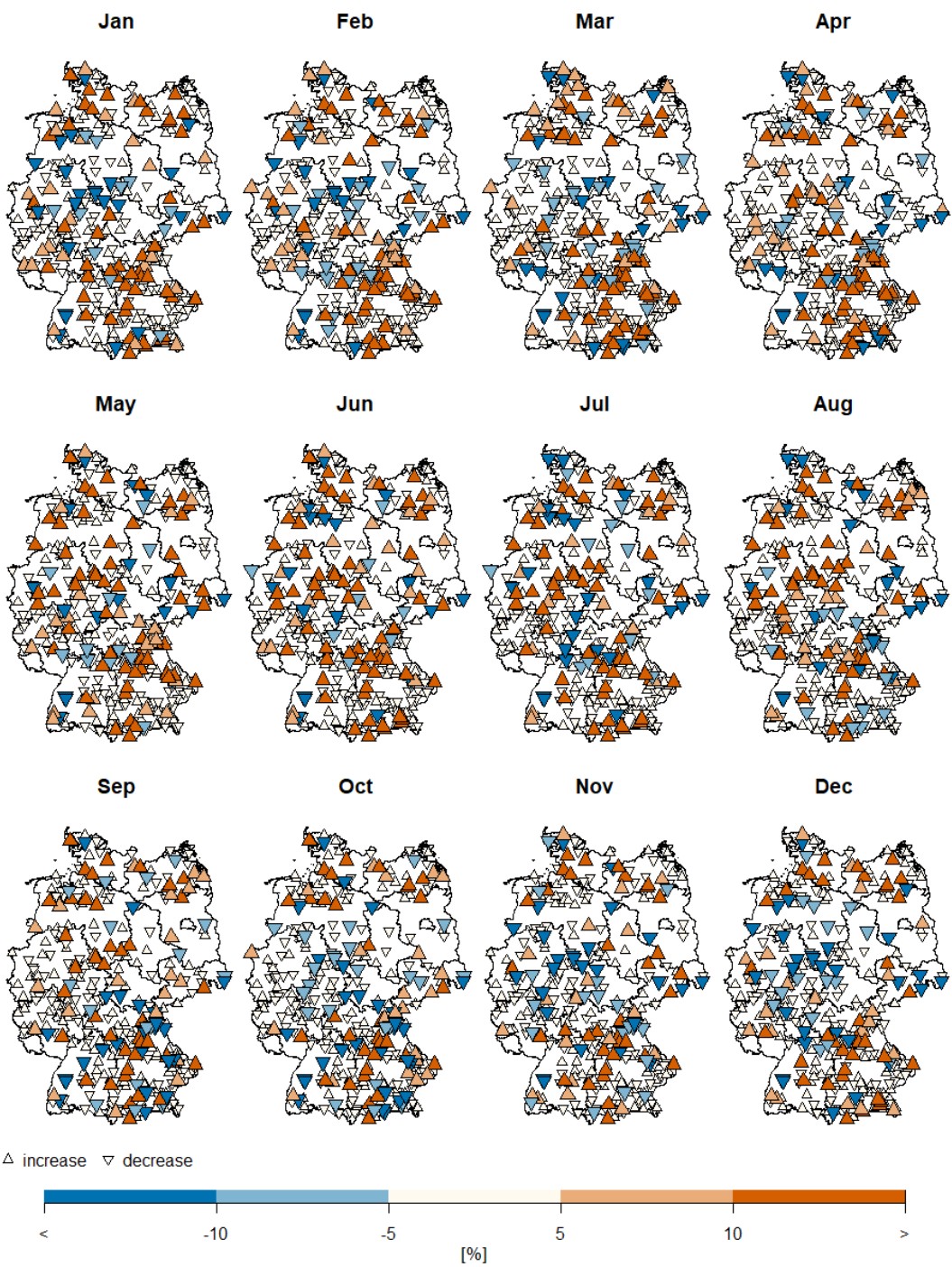

**Figure B2.** Relative change from 1941 to 2021 for the 100-year return level (non-exceedance probability of $p = 0.99$) for 338 stations. Increases / decreases are marked with pointing up triangles / pointing down triangles. Minor changes ($< 5$ %) remain uncolored with small symbols, stronger increases / decreases are highlighted in orange / blue.

once every 100 years, in 2050 the same event might be exceeded on average once in 90 years. The changing return period (or exceedance probability) is an obstacle for engeneering applications. One solution is given by the design-life level, which accounts for the time when the hydraulic system will be build and the service life of the system, called the design-life period. While the design-life period should be very long for dike design (e.g. 10.000 years in Netherlands (Botzen et al., 2009)), the service life of a rain gutter is much shorter.

The design-life level $r_p$ can be obtained by numerically optimizing the equation:

$$\prod_{i=1}^{I} G_i(r_p) = p \tag{C1}$$

with $G_i$ being the Generalized Extreme Value distribution for year $i$, $p$ the non-exceedance probability and $I$ the design-life period. This approach assumes independent maxima. The design-life level is stated as $T_1$ - $T_2$ *(1-p)% extreme level* with $T_1/T_2$ indicating the start / end of the design-life period. To calculate future design-life levels, we use the seasonal-interannual and

the seasonal-only model to extrapolate the parameters of the GEV for the month July at the station *Rain am Lech* until 2051 (Fig. C1). With Eq. C1, the 2022-2051 1% extreme precipitation level ($I = 30$ ,$p = 0.99$) for the month July at *Rain am Lech* obtained with the seasonal-interannual model equals to 161.4 mm/day. In other words, there is a 1 in 100 risk that the largest daily precipitation event during 2022 - 2051 will be higher than 161.4 mm/day. The 2022-2051 1% extreme precipitation level for the seasonal-only approach is 132.5 mm/day. If the detected trend at Rain am Lech continues for the years 2022 - 2051,

as assumed here, the seasonal-only approach will lead to underestimated risks and the designed risk adaptation system will be strained beyond its planning purpose.

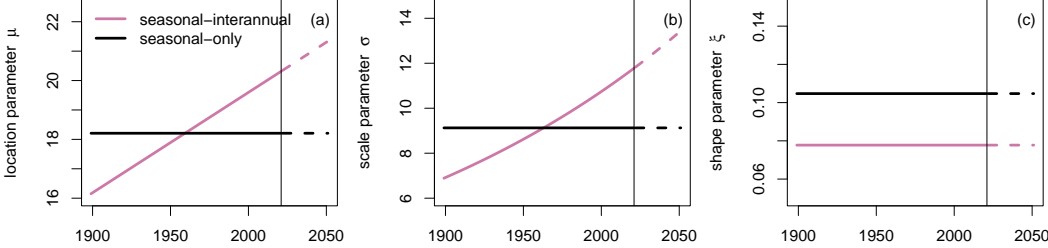

**Figure C1.** Estimated parameter for location $\mu$ (a), scale $\sigma$ (b) and shape $\xi$ (c) at the example station *Rain am Lech* for the month July using a seasonal-interannual model (pink) and a seasonal-only model (black). Additionally to the estimates for the observation period (solid line), extrapolated values since 2022 are also illustrated (dashed lines).

*Author contributions.* MP, HR and UU designed the study concepts and methodology. MP conducted the analysis, generated the results and wrote the first draft. All authors contributed to writing the manuscript and approved the final version.

*Competing interests.* Some authors are members of the editorial board of NHESS. The peer-review process was guided by an independent
editor, and the authors have also no other competing interests to declare.

*Acknowledgements.* This study has been partially funded by the Deutsche Forschungsgemeinschaft (DFG) within the research training programme *NatRiskChange* GRK 2043/1 at Potsdam University. Additionally, the authors thank the National Climate Data Center of the German Weather Service (DWD) for providing and maintaining the precipitation datasets (https://opendata.dwd.de/). We thank Oscar Jurado de Larios and Felix Fauer for proof-reading and Theano Iliopoulou and the anonymous referee for their careful reading of our manuscript
and their constructive comments. The analysis was carried out using R, an environment for statistical computing and graphics Team (2016), based on the VGAM package Yee (2015a).

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
