# Peer review of "Interannual variations in the seasonal cyle of extreme precipitation in Germany and the response to climate change"

_Natural Hazards and Earth System Sciences, 2023_

## Referee Comment (RC2)

This work presents a non-stationary methodology to model seasonal and interannual variability of monthly maxima of daily precipitation, based on the Generalized Extreme Value distribution, applied to 519 stations in Germany. The subject is interesting, and the manuscript is very well structured, with appropriately designed analyses. Although the work is clearly presented, it follows a highly algorithmic approach which at times is difficult to follow. My remarks are mainly focused on the hydrological and practical relevance of the methodology particularly with respect to the nonstationary modelling of the interannual variations.

**Major comments**

-Although the study of seasonality through cyclo-stationary models is well established in the literature, modelling of interannual variations using nonstationary models is not as common. A possible reason might be that interannual variations do not have a well-understood physical basis, which is a theoretical requirement in order to rigorously apply nonstationary models (see e.g. Montanari and Koutsoyiannis, 2014). Rainfall interannual variations are usually irregular and linked to rainfall's natural variability, which is typically quantified and modelled with stochastic and stationary approaches (e.g. Iliopoulou et al. 2018; Iliopoulou and Koutsoyiannis, 2019). In this respect, I think it would be beneficial to expand the discussion in the Introduction on the rationale and scope of using a nonstationary approach to model interannual variability.

-A similar question relates to how such a method could be applied in practice. For instance, could the authors provide an example of a seasonal-interannual nonstationary EV modelling for a selected station compared to an application of their seasonal-only approach?

-Regarding spatial consistency, Figure 6 suggests that stations with interannual components do not follow a specific spatial pattern, which could be potentially indicative of a physical mechanism, but rather show a large spatial variability, which might be indicative of a large uncertainty involved in the identification of these variations. How do the authors explain this spatial variability? Does the proposed nonstationary approach allow accounting for uncertainty in parameter fitting?

**Minor comments**

-Conclusions: It would also be interesting to note here the percentage of stations favoring a seasonal-only variation of the shape parameter.

-Please explain subscripts $i, j$ in Equation (3).

-Line 110: typo 'interactions'

**References**

Iliopoulou, T., Papalexiou, S.M., Markonis, Y. and Koutsoyiannis, D., 2018. Revisiting long-range dependence in annual precipitation. *Journal of Hydrology*, *556*, pp.891-900.

Iliopoulou, T. and Koutsoyiannis, D., 2019. Revealing hidden persistence in maximum rainfall records. *Hydrological Sciences Journal*, *64*(14), pp.1673-1689.

Montanari, A. and Koutsoyiannis, D., 2014. Modeling and mitigating natural hazards: Stationarity is immortal!. *Water Resources Research*, *50*(12), pp.9748-9756.

---

## Author Comment (AC1)

**Reply to Reviewer 1 for manuscript "Interannual variations in the seasonal cyle of extreme precipitation in Germany and the response to climate change"**

Madlen Peter, Henning W. Rust and Uwe Ulbrich

Institute of Meteorology, Freie Universität Berlin, Germany

August 30, 2023

**Reviewer 1:** This manuscript presents an analysis of the seasonal and interannual variations of extreme precipitation at stations in Germany, employing a non-stationary block maxima approach. Additionally, it investigates the impact of climate change on the seasonal cycle of extreme precipitation in Germany, which is a crucial topic in climate change research. The paper is well-structured and complemented by visually appealing figures. However, there are several issues that require attention and improvement before this work can be considered for publication.

**Main comments**

- **Reviewer 1:** For introduction, according to the objective of the paper, it is important to address what previous studies have specifically accomplished, identify the existing gap or problem in the research, and emphasize why this problem is of significant concern. It is crucial to provide clarity on these aspects before describing the approach or research you intend to use in your study. For example, the third paragraph of the introduction discusses previous analyses conducted on extreme precipitation in Germany across different seasons. "Analyses of extreme precipitation in Germany for different seasons has already been done (Zolina et al., 2008; Łupikasza, 2017; Fischer et al., 2018; Zeder and Fischer, 2020; Ulrich et al., 2021)." More details of what previous studies have done are needed before you introduce the two main new aspects you will do in this study. In addition, the second question is "RQ2 How important is a flexible shape parameter to reflect recorded variations?". However, you did not add any descriptions or previous studies about shape parameter in the introduction. Therefore, My suggestion is to rewrite the introduction.

**Answer:** We have reworked the introduction and added some detailed information about previous studies: "Zolina et al. (2008) and Łupikasza (2017) analysed quantiles of daily precipitation sums separately for the seasons DJF, MAM, JJA and SON, while Fischer et al. (2018, 2019) used available data more efficiently by modelling monthly maxima of daily precipitation sums for all months simultaneously. This approach has been proven to lead to more robust and reliable results than considering months separately. Ulrich et al. (2021) extended this method by including different durations to efficiently estimate intensity-duration-frequency curves. Furthermore, Zeder and Fischer (2020) analysed the effect of climate change on seasonal extreme precipitation and found a positive connection to the north-hemispheric temperature rise. In our approach we combine the simultaneous modelling of available data for all months with interannual variations, thus accounting for potential changes of the seasonality due to climate change and natural variability."

Additionally, we have added some sentences about the importance of the shape parameter: "The goal of this paper is to assess the performance of the seasonal-interannual modelling with a special attention to a flexible shape parameter $\xi$. This parameter is difficult to estimate as it interferes with the scale parameter (Ribereau et al., 2011) and requires long records for reliable results (Papalexiou and Koutsoyiannis, 2013). Nevertheless, it describes the behaviour of the very rare events and consequently plays an important role for assessing extreme precipitation changes."

- **Reviewer 1:** For method: return level, the return period T can be written as $T = \mu/(1 - p)$, where, p is the non-exceedance probability. $\mu$ is the mean interarrival time between two successive events, which is defined as one divided by the number extreme events per year. When considering annual maxima, $\mu$ corresponds to 1 year. However, in your study, when calculating the return period T, are we utilizing the annual or monthly maximum or non-exceedance probabilities? If we are using the monthly maximum time series or non-exceedance probabilities, $\mu$ should not be equal to 1.

  **Answer:** We calculate the January (or February or...) return levels expressed in frequency per January (or February or ... ), for example, the one in ten Januarys return level (10-January return level). This should not be mistaken with the annual return levels obtained with annual maxima. We have added a paragraph to Section 3.5 to clarifiy which return levels are considered in our manuscript: "As we consider monthly maxima we calculate as well monthly return levels. Similar to e.g. 100-year return levels obtained with annual maxima, we determine the 100-January return levels, the 100-February return levels, and so on. In the following we state them as monthly 100-year return levels instead of naming respective months. This should not be confused with annual return levels. However, they can be calculated as well with monthly maxima leading to

more accurate and reliable annual results (Maraun et al., 2009; Fischer et al., 2018)."

- **Reviewer 1:** In addition, when applying the GEV to the monthly maximum, if two extreme events occur on the last day of the month and the first day of the next month, these two events are often treated as a single individual event. When applying the GEV to non-exceedance probabilities, precipitation occurrences are highly clustered in time and space. Therefore, the independence of the extreme values should be taken into account prior to modeling.

  **Answer:** Indeed, it could happen that precipitation maxima of two successive months belong to the same precipitation event and maxima are not completely independent in time. However, for our dataset, this is the case for only about 0.6% of the data. Here, the temporal dependence is neglected and independence is assumed. We added to Section 3.1 the sentence: "This requires independent block maxima of successive months. However, this assumption can be violated if two monthly maxima belong to the same precipitation event, e.g. if one maximum occurs at the end of the month and the second one at the beginning of the next month. For the given records, about 0.6% of the monthly maxima have been registered at successive days. Since this fraction is low, we neglect temporal dependances and assume independent monthly maxima."

- **Reviewer 1:** The fitted return period distribution may exhibit uncertainties due to the limited sample size of the data. The short time period of the datasets may introduce uncertainty in the distribution model fitting. Therefore, for the question "RQ1: Can a model with interannual variations better represent the observations than a seasonal-only model?" how can you distinguish the difference or bias from the uncertainty in distribution model fitting or from the model with or without interannual variations? As shown in the paper "the total QSS for different non-exceedance probabilities (return periods). Skill is positive but small$<= 2\%$, increasing with non-exceedance probability (return period)." The larger bias for higher return period is very likely caused by large uncertainties for higher return period in model fitting.

  **Answer:** Indeed, quantile estimates for higher non-exceedance probabilites are related with higher uncertainties. This is also reflected in the QSS as you stated. Thus, we have mentioned that evaluations for return levels with a return period larger than the observation period need to be treaten cautiously. We have written in the manuscript: "The latter has to be interpreted with care as there are very few observations in the range of the upper quantiles. Return levels with a return period higher than the time range of the data should be treated cautiously, since the quantile score can not reasonably evaluate those values (Fauer and Rust, 2023)." Here, this is only the case for non-exceedance probabilities of 0.99 and 0.995 (return period of 100 and 200 years). We have added to the text:

"As we consider for each station at least 80 years of observations, this only matters for non-exceedance probabilities (return periods) of 0.99 and 0.995 (100 and 200 years).". Uncertainties exist in both models, a seasonal-interannual model and a seasonal-only model. However, we have analysed in a cross-validated approach, that the interannually varying model is beneficial. Here, the outcome "Skill is positive but small $\lesssim 2\%\%$" refers to the overall investigation summarising the skill for all months and stations. However, skill is more pronounced for single stations and different months, which is illustrated in more detail in Fig. 7, Fig. 8 and Fig. 10.

**Other comments**

- **Reviewer 1:** Lines 21-25: Please maintain consistency in the usage of terminology such as 'heavy rain' or 'heavy precipitation' throughout the entire

  **Answer:** Many thanks for the hint. We have changed it to 'heavy precipitation' for the entire document.

- **Reviewer 1:** Line 77-79, "The four stations Krümmel (1899-01-01 until 2021-12-31), Mühlhausen / Oberpfalz-Weiherdorf (1931-01-01 until 2021-12-31), Rain am Lech (1899-01-01 until 2021-12-31) and Wesertal-Lippoldsberg (1931-01-01 until 2021-12-31) are highlighted in Fig. 1 and will be discussed exemplarily in this study". Why do you choose these four stations? It would be beneficial to include a brief introduction explaining the reasons behind selecting these stations. Although you provide more details about the stepwise selection process in Section 4.1, adding an introductory explanation would provide context for the readers.

  **Answer:** We have added the sentence: "We have selected these stations as they are characterised by different changes in seasonality (see Sec. 7) represented by divergent model setups (see Sec. 4.1). Additionally, their interannual changes are more pronounced than for other stations."

- **Reviewer 1:** The sample size of the data in model fitting. For example, for figure 7 and figure 8, the GEV was applied to each station, especially for each month of each station, how many extreme values (sample) are you collected for each station for each month? Are the number of samples enough for distribution model fitting?

  **Answer:** Since we consider stations with at least 80 years of observations, there should be at least 80 data points for each station and month. However, missing values within the observation period are allowed such that less than 80 data points might be available. The minimum number of maxima for one month and station is 78. We have checked exemplarily the stationary GEV for this month by using model diagnose (qq-plot, pp-plot, return level plot and histogram), revealing that a sample size of 78 data points is sufficiently large enough for model fitting.

- **Reviewer 1:** Figure 12, it is better to add a,b,c,d into the each figure.

  **Answer:** added

- **Reviewer 1:** In Section 7, "Impact of climate change on the seasonality of extreme precipitation," it is important to note that the trend of the time series is significantly influenced by the chosen start year. Although the study mentions comparing the time period from 1941 to 2021, where all stations have data, the start year appears to be different in Figure 15. Could you please provide further clarification on the discrepancy?

  **Answer:** We describe the problem of calculating linear trends for fixed time periods in detail in the first paragraph of Section 7 and in Appendix A. To highlight that this is not done for the analyses of the example stations, we have added to the paragraph "We compare the time period from 1941 to 2021 where all stations have data. Note that estimating linear trends for fixed (and short) periods of time can yield very different results depending on the considered time period due to decadal variability. Thus, the trend estimates presented here for the given time period serve as a rough indicator for climate change effects; for a more detailed analysis the whole datasets should be taken into account for each station" the following part: ", which is done in Fig. 15.".

- **Reviewer 1:** In section 7, the return period was calculated for each station for each year? is the sample size enough for model fitting?

  **Answer:** The non-stationary model, which considers maxima of all months and years simultaneously, allows for varying return within the year and throughout the years. Since we only consider stations with at least 80 years of observations, a minimum of 960 data points (80 times 12) are available for each station which is sufficient for model fitting. Due to the interannual component in modelling we are able to estimate different return levels for each year.

---

## Author Comment (AC2)

**Reply to Reviewer 2 for manuscript "Interannual variations in the seasonal cyle of extreme precipitation in Germany and the response to climate change"**

Madlen Peter, Henning W. Rust and Uwe Ulbrich

Institute of Meteorology, Freie Universität Berlin, Germany

August 31, 2023

**Reviewer 2:** This work presents a non-stationary methodology to model seasonal and interannual variability of monthly maxima of daily precipitation, based on the Generalized Extreme Value distribution, applied to 519 stations in Germany. The subject is interesting, and the manuscript is very well structured, with appropriately designed analyses. Although the work is clearly presented, it follows a highly algorithmic approach which at times is difficult to follow. My remarks are mainly focused on the hydrological and practical relevance of the methodology particularly with respect to the nonstationary modelling of the interannual variations.

**Major comments**

- **Reviewer 2:** Although the study of seasonality through cyclo-stationary models is well established in the literature, modelling of interannual variations using nonstationary models is not as common. A possible reason might be that interannual variations do not have a well-understood physical basis, which is a theoretical requirement in order to rigorously apply nonstationary models (see e.g. Montanari and Koutsoyiannis, 2014). Rainfall interannual variations are usually irregular and linked to rainfall's natural variability, which is typically quantified and modelled with stochastic and stationary approaches (e.g. Iliopoulou et al. 2018; Iliopoulou and Koutsoyiannis, 2019). In this respect, I think it would be beneficial to expand the discussion in the Introduction on the rationale and scope of using a nonstationary approach to model interannual variability.

  **Answer:** In our analysis, we do not consider differences between successive years as interannual variations, but rather the trend, which could be non-linear as well. We added the sentence to the introduction: "Here,

we point out that when referring to interannual variations, we are not addressing differences between successive years, but rather the trend over the entire observation period, which could be potentially non-linear." Additionally, we have added some sentences in the introduction to clarify the aim of using a nonstationary approach for these interannual changes: "Interannual variations in precipitation have been shown to be associated with its natural variability (e.g. Willems, 2013), increased air temperatures (Trenberth et al.,2003; Westra et al., 2013, 2014) and other effects influencing large-scale atmospheric circulations and precipitation characteristics (Pinto et al., 2007, 2009; Davini and d'Andrea, 2020; Detring et al., 2021). Most of these effects are highly non-linear and their roles are difficult to quantify. Here, we use time as proxy to combine those different unknown effects." Furthermore, we have included a sentence to highlight that our seasonal-interannual model is able to reflect linear changes and natural variability (higher-ordered polynomials): "Here, we use Legendre polynomials up to an order of five to describe the variations across years. This enables on the one hand the reflection of changes potentially associated with climate change and on the other hand allows for modelling of natural variability in extreme precipitation."

- **Reviewer 2:** A similar question relates to how such a method could be applied in practice. For instance, could the authors provide an example of a seasonal-interannual nonstationary EV modelling for a selected station compared to an application of their seasonal-only approach?

  **Answer:** A seasonal consideration of maxima in extreme value statistics provides additional information (monthly resolved return levels) and improves the quality of the return levels, as analysed in Fischer et al. (2018, 2019). An interannual consideration fosters on the one hand the knowledge about how seasonality in extreme precipitation has changed and on the other hand it enables more adabted concepts for engeneering purposes. To demonstrate the advantages, we have added the 100-year return levels for the seasonal-only model to Fig. 15 and wrote one sentence per example station to the main text. For *Rain am Lech*: "Considering the 100-year return levels of the seasonal-only model demonstrates that a non-interannual approach leads to highly underestimated values especially for the first record decades." For *Wesertal-Lippoldsberg*: "The model verification (Fig. 10) confirms that a model with a changing seasonal cycle better represents the data observed in summer for return periods of 10 to 50 years, while the 100- and 200-year return levels are strongly overestimated with respect to the observations, especially for the most recent decades. In constrast, the seasonal-interannual model is more beneficial for estimating winterly return levels with return periods longer than 30 years. These characteristics can be seen as well by comparing the 100-year return levels of the seasonal-interannual model with those of the seasonal-only model." For *Mühlhausen / Oberpfalz-Weihersdorf*: "Although differences between the 100-year return levels of the seasonal-only and the

seasonal-interannual model are not very pronounced, the shift from late summer to early summer, which might be continued in future, cannot be detected with the non-interannual approach." For *Krümmel*: "For this example, the seasonal-only approach applied to the whole record might be beneficial in terms of long-term risk assessment and hydraulic design since natural variability does not play a key role for longer planning horizons. However, for short- to mid-term risk assessment, e.g. for agriculture or tourism sector, the natural variability might be of relevance."

Furthermore, we have included a section to the discussion in 8.3 pointing out a potential application of an interannual modelling approach for quantifying and communicating environmental risks in a changing world by calculating design-life levels. In addition, we added in Appendix C a brief explanation and calculation of the design-life level. We wrote in 8.3: "A possible application of the presented seasonal-interannual approach in the field of risk adaptation could be realised by calculating design-life levels. This concept has been introduced by Rootzén and Katz (2013) and widely applied in research and risk management (e.g. Thomson et al., 2015; Mondal and Daniel, 2019; Xu et al., 2019; Byun and Hamlet, 2020). The design-life level is a measure for quantifying and communicating environmental risks in a changing climate accounting for the service life of a system (design-life period, e.g. 30 years) and the time, when the system will be installed (e.g. in 2025). Due to changing extreme precipitation characteristics, the 2025-2055 1% design-life level could be different from the 2055-2085 1% design-life level. More detailed explanations and example calculations can be found in Appendix C. The seasonal-interannual modelling approach can be used to calculate future seasonal design-life levels either by extrapolating past climate trends or by an application to outputs from climate projections. Since for risk adaptation in an engineering context annual design-life levels are more beneficial then seasonal ones, the same methodological concept can be applied to obtain annual values out of a seasonal modelling approach (Maraun et al., 2009; Fischer et al., 2018)."

Appendix C has been added: "According to Rootzén and Katz (2013), the design-life level is a measure to quantify risks for engeneering design purposes in a changing climate. This measure can be regarded as a logical extention of the return level approach which can only be meaningfully interpreted in a stationary setting. For example, a 100-year return level of extreme precipitation is the value which is expected to be exceeded in mean once in hundred years. Due to changing climate, an event can occur in 2023 once every 100 years, in 2050 the same event might be exceeded on average once in 90 years. The changing return period (or exceedance probability) is an obstacle for engeneering applications. One solution is given by the design-life level, which accounts for the time when the hydraulic system will be build and the service life of the system, called the design-life period. While the design-life period should be very long for

dike design (e.g. 10.000 years in Netherlands (Botzen et al., 2009)), the service life of a rain gutter is much shorter. The design-life level $r_p$ can be obtained by numerically optimizing the equation:

$$\prod_{i=1}^{I} G_i(r_p) = p \qquad (1)$$

with $G_i$ being the Generalized Extreme Value distribution for year $i$, $p$ the non-exceedance probability and $I$ the design-life period. This approach assumes independent maxima. The design-life level is stated as $T_1$ - $T_2$ (1-p)% extreme level with $T_1/T_2$ indicating the start / end of the design-life period.

[Figure]

Figure 1: Estimated parameter for location $\mu$ (a), scale $\sigma$ (b) and shape $\xi$ (c) at the example station *Rain am Lech* for the month July using a seasonal-interannual model (pink) and a seasonal-only model (black). Additionally to the estimates for the observation period (solid line), extrapolated values since 2022 are also illustrated (dashed lines).

To calculate future design-life levels, we use the seasonal-interannual and the seasonal-only model to extrapolate the parameters of the GEV for the month July at the station *Rain am Lech* until 2051 (Fig. C1). With Eq. 1, the 2022-2051 1% extreme precipitation level ($I = 30$, $p = 0.99$) for the month July at *Rain am Lech* obtained with the seasonal-interannual model equals to 161.4 mm/day. In other words, there is a 1 in 100 risk that the largest daily precipitation event during 2022 - 2051 will be higher than 161.4 mm/day. The 2022-2051 1% extreme precipitation level for the seasonal-only approach is 132.5 mm/day. If the detected trend at *Rain am Lech* continues for the years 2022 - 2051, as assumed here, the seasonal-only approach will lead to underestimated risks and the designed risk adaptation system will be strained beyond its planning purpose.

- **Reviewer 2:** Regarding spatial consistency, Figure 6 suggests that stations with interannual components do not follow a specific spatial pattern, which could be potentially indicative of a physical mechanism, but rather show a large spatial variability, which might be indicative of a large uncertainty involved in the identification of these variations. How do the authors explain this spatial variability? Does the proposed nonstationary approach allow accounting for uncertainty in parameter fitting?

**Answer:** It is true that there is spatial variability in the selected models of neighbouring stations. There are several reasons:

1. Extreme precipitation is partly very small-scaled and one station could be affected by a heavy precipitation event while at a neibouring station it does not rain at all. A common problem in extreme value analysis is that very rare events could have large influence on the extreme value distribution especially if a short record is available.

2. The GEV parameter estimates interfere (Ribereau et al.,2011). For example, if the shape parameter is slightly increased the scale parameter will be adapted as well, such that the distribution will be fitted suitably to the data. Thus, similar distributions could be described with different parameters.

3. The model selection procedure select one suitable model. However, a different model could be even as good as the selected one. In each iteration of the stepwise-forward selection the BIC of every possible covariate is obtained and the model with the lowest BIC is selected. However, the difference in BIC between the best and the second best candidate could be negligible.

Spatial variability in the selected models is present. However, the synergy of all three GEV parameters and different covariates can model similar characteristics in extreme precipitation at neibouring stations, which can be seen for the estimated return levels. Nevertheless, common spatial characteristics in the selected interannual covariates can be detected, like described in the manuscript. We have added a sentence about the spatial variability in Section 4.1: "It can be seen that the selected interannual covariates are partly very variable in space. This can be explained by 1) a large spatial variability in extreme precipitation due to small-scaled very intense events and 2) the model selection procedure, which chooses one suitable model, even if other models are comparably appropriate. However, common characteristics can be detected:"

Uncertainty in parameter estimation can be taken into account by calculating confidence intervals, e.g. using the delta method (Coles, 2001). Since the aim of our investigation was to analyse whether interannual changes in seasonal extreme precipitation in Germany can be detected with a nonstationary approach, we have refrained from integrating confidence intervals into our investigation. As we believe that uncertainty in return level estimates are crucial, we added a paragraph to the outlook section in 8.3: "In our investigation we consider return level estimates. However, analysing their uncertainties are crucial. For further investigations, confidence intervals, e.g. calculated with the delta method (Coles, 2001), should be taken into account. A comparison of uncertainties evolved by the seasonal-interannual model and those of a seasonal-only model could deepen the investigation if interannual models are beneficial for risk assessment or

if the changing return levels are rather within the uncertainty range of non-interannually varying return levels."

**Minor comments**

- **Reviewer 2:** Conclusions: It would also be interesting to note here the percentage of stations favoring a seasonal-only variation of the shape parameter. **Answer:** We have added a sentence in the maintext in Section 6: "Most of the stations (106 / 178, about 60 %) are represented by a model including seasonal variations, whereby many of them (92 /106 stations) do not favor an interannnually varying shape parameter at all." Additionally, we added in the conclusion: "The BIC based model selection strategy favours a flexible shape for 178 / 519 stations (about 34%), whereby about 52% (92/178) of these records prefer a seasonal-only component. For the remaining stations with variable $\xi$, an interannually changing seasonality occurs more often than the direct interannual variations.

- **Reviewer 2:** Please explain subscripts i, j in Equation (3). **Answer:** We have added to the text: "$\mu_0$ denotes the constant intercept (offset), the second term the direct effects of a covariate $X_i$, e.g. seasonal or interannual, and the third term the intercations between different dimensions (indicated by $i$ and $j$), e.g. seasonal and interannual."

- **Reviewer 2:** Line 110: typo 'interactions' **Answer:** changed